# THOUGHT PROPAGATION:
# AN ANALOGICAL APPROACH TO COMPLEX REASONING WITH LARGE LANGUAGE MODELS

**Junchi Yu & Ran He** *
MAIS& CRIPAC, Institute of Automation
Chinese Academy of Sciences
University of Chinese Academy of Sciences
Beijing, China
`yujunchi2019@ia.ac.cn,`
`rhe@nlpr.ia.ac.cn`

**Rex Ying**
Department of Computer Sciences
Yale University
New Haven, USA
`rex.ying@yale.edu`

## ABSTRACT

Large Language Models (LLMs) have achieved remarkable success in reasoning tasks with the development of prompting methods. However, existing prompting approaches cannot reuse insights of solving similar problems and suffer from accumulated errors in multi-step reasoning, since they prompt LLMs to reason *from scratch*. To address these issues, we propose **Thought Propagation** (**TP**), which explores the analogous problems and leverages their solutions to enhance the complex reasoning ability of LLMs. These analogous problems are related to the input one, with reusable solutions and problem-solving strategies. Thus, it is promising to propagate insights of solving previous analogous problems to inspire new problem-solving. To achieve this, TP first prompts LLMs to propose and solve a set of analogous problems that are related to the input one. Then, TP reuses the results of analogous problems to directly yield a new solution or derive a knowledge-intensive plan for execution to amend the initial solution obtained from scratch. TP is compatible with existing prompting approaches, allowing plug-and-play generalization and enhancement in a wide range of tasks without much labor in task-specific prompt engineering. Experiments across three challenging tasks demonstrate TP enjoys a substantial improvement over the baselines by an average of 12% absolute increase in finding the optimal solutions in Shortest-path Reasoning, 13% improvement of human preference in Creative Writing, and 15% enhancement in the task completion rate of LLM-Agent Planning. Code is available on `https://github.com/Samyu0304/thought-propagation`.

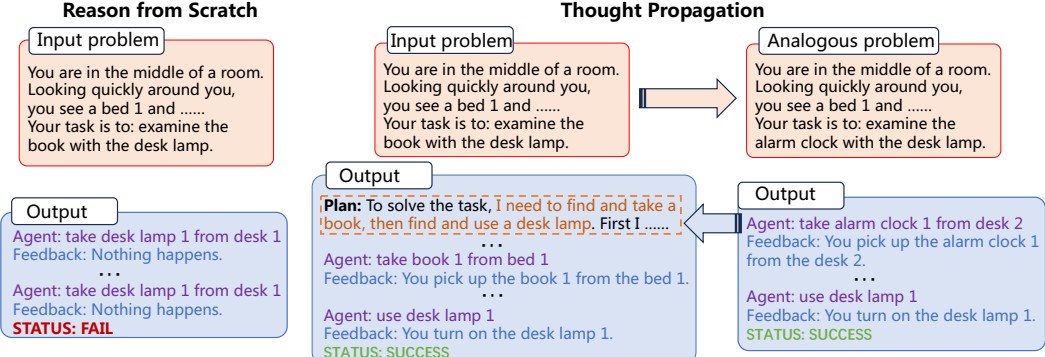

Figure 1: Reasoning from scratch cannot reuse insight of solving similar problems and suffers from accumulated errors in intermediate reasoning stages. Thought Propagation explores analogous problems that are related to the input one and derives insights from solutions to analogous problems.

---

* Corresponding Author.

# 1 INTRODUCTION

The scaling-up Large Language Models (LLMs) (OpenAI, 2023) have achieved notable success in logical (Khot et al., 2023), arithmetic (Wei et al., 2022), and commonsense (Liu et al., 2021) reasoning with the development of prompting methods (Qiao et al., 2022). The early works employ few-shot input and output exemplars to prompt the LLM to perform simple reasoning (Brown et al., 2020). Recent methods decompose the complex reasoning process into intermediate reasoning steps to enable LLMs with multi-step reasoning abilities (Wei et al., 2022; Wang et al., 2023d).

Although many efforts are made to improve complex reasoning with LLMs by crafted prompt design (Zhang et al., 2022b), delicate decomposition (Khot et al., 2023; Zhou et al., 2023), and advanced searching scheme (Yao et al., 2023), these methods prompt the LLM to reason *from scratch*. This scheme is problematic to complex reasoning for two reasons. First, reasoning from scratch cannot reuse the insights of solving similar problems (Hall, 1989). Using such insights as prior knowledge can ease the difficulty of solving complex problems and develop new solutions (Carbonell, 1985). One can further assess new solutions with rough ones obtained from scratch and yield refined solutions. Second, reasoning from scratch is sensitive to the errors made in intermediate stages when facing tasks involving multi-step reasoning. As shown in Figure 1, these errors will accumulate as they misguide the searching and planning afterward, which eventually leads to invalid reasoning outcome (Dziri et al., 2023; Yao et al., 2022). These two challenges motivate us to develop an alternative framework for LLM reasoning to amend the limitations of reasoning from scratch.

The study of human cognition presents a promising way to amend the limitations of reasoning from scratch, primarily through the application of analogy (Bartha, 2013). The analogy highlights the occurrence of entities' relationship in the form of "A is to B what C is to D". By discerning and applying such analogical reasoning, humans can stand on the shoulder of existing knowledge to pioneer new concepts in novel domains. A compelling historical example is the discovery of Coulomb's Law, which can be traced back to the analogy drawn between gravitational forces governing celestial bodies and electrical forces acting upon charged particles (Priestley, 1775). Such a framework has been recently proven to be efficient in relational reasoning between entities on knowledge graphs (Zhang et al., 2022a; Yuan et al., 2023). However, a general framework of harnessing analogies among problems to facilitate LLM reasoning, to the best of our knowledge, is yet to be explored.

Hence, we advance the traditional analogical reasoning and propose a novel ***Thought Propagation* (TP)** framework to amend existing reasoning-from-scratch methods and enhance the complex reasoning ability of LLMs. Given an input problem, TP first prompts LLMs to propose a set of analogous problems that are related to the input one. Then, the input problem with its analogous counterpart is solved by existing prompting approaches such as Chain-of-Thought (CoT) (Wei et al., 2022). An aggregation module further aggregates the solutions from these analogous problems, facilitating input problem-solving through two distinct avenues. First, this module reuses the solutions derived from analogous problems to generate a new solution to the input problem. The aggregation module assesses the new solution produced by the analogical approach with the initial one obtained from scratch to output a refined result for the input problem. Second, this module compares the input problem with its analogous counterparts and devises high-level plans based on the results of analogous problems. These plans are then executed by the LLM to rectify its intermediate reasoning steps when addressing the input problem. TP is compatible with existing approaches, allowing plug-and-play generalization and enhancement to various tasks ranging from optimization problems to autonomous agent planning. Thus, it reduces intensive labor in task-specific prompt engineering.

We test the proposed method on three tasks, including Shortest-path Reasoning, Creative Writing, and LLM-Agent Planning. These tasks require searching over graph-structure data, open-ended writing, and long-trial planning, which challenge existing methods for LLM reasoning. Experimental results show that Thought Propagation can generalize to a wide range of different reasoning tasks and enjoys superior performances on all of them.

# 2 RELATED WORK

**Graph Neural Network**. Graph neural networks (GNNs) are expressive network backbones of deep graph learning due to their inductive bias on graph-structured data (Hamilton et al., 2017; Kipf & Welling, 2017). The expressiveness of GNNs can improve by discovering the task-related

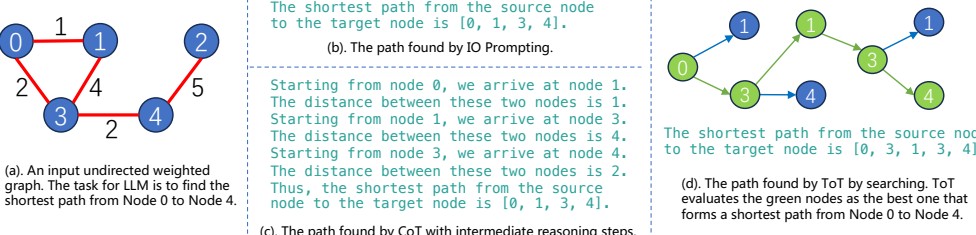

Figure 2: An example of existing methods on shortest path reasoning task. Although the graph in (a) is quite simple, these methods only prompt the LLM to find the sub-optimal solutions (b,c), and even repeatedly visit intermediate nodes (d), due to reasoning from scratch.

structures on graphs (Yu et al., 2021a;b; Sun et al., 2021; Li et al., 2023). Recent works incorporate parameterized GNNs with Large Language Models (LLMs) for graph-related tasks, such as graph explainability (He et al., 2023), classification (Chen et al., 2023c; Qian et al., 2023), and question answering (Jiang et al., 2023). Differently, our work aims to improve the general reasoning ability of LLMs using problem analogy without fine-tuning.

**Analogical Reasoning**. The analogical reasoning has been applied to visual reasoning (Małkiński & Mańdziuk, 2022), natural language reasoning (Chen et al., 2022; Sultan & Shahaf, 2022), and knowledge graph reasoning (Zhang et al., 2022a). These methods train parameterized neural networks to perform relational reasoning between entities. Early attempts have shown LLMs can do analogical reasoning just like humans by case study (Webb et al., 2023). Recent works explore analogy generation and analogy reasoning with knowledge graphs on LLMs (Yuan et al., 2023; Bhavya et al., 2022; 2023). However, they focus on different applications instead of general reasoning problems. Moreover, they rely on large-scale external knowledge bases to store entity relationships to perform analogical reasoning, which is expensive for general reasoning tasks. Thus, a general analogical approach for LLM complex reasoning on general tasks is still in its vacuum.

**Prompt-based Large Language Model Reasoning**. Originally designed for text generation, the Large Language Models (LLMs) have succeeded in many applications with prompting methods (Liu et al., 2023b; Zhao et al., 2023). Early methods employ input and output (IO) prompting that appends pairs of problems and solutions exemplars on top of the input problem (Brown et al., 2020). Recent methods decompose the complex reasoning process into intermediate reasoning steps. They use multi-step prompting (Wei et al., 2022; Wang et al., 2023d; Zhang et al., 2022b) or recursive problem decomposition (Zhou et al., 2023; Khot et al., 2023) to enable multi-step LLMs reasoning. Others design searching schemes for LLMs (Yao et al., 2023; Besta et al., 2023). However, they solve each problem from scratch. Thus, they cannot reuse the insights of solving similar problems. Moreover, they suffer from accumulated errors in intermediate reasoning steps.

## 3 PRELIMINARIES

Denote the reasoning problem and the solution as $\mathbf{p} \in \mathcal{P}$ and $\mathbf{s} \in \mathcal{S}$. $\mathcal{P}$ and $\mathcal{S}$ are the problem and solution space. Let the LLM be $f_\theta$ where $\theta$ denotes model weights.

**IO Prompting**. IO prompting (Brown et al., 2020) uses task descriptions and few-shot pairs of Input and Output (IO) prompting demonstrations to assist LLMs in reasoning to solve the input problem $\mathbf{p}$ by $\mathbf{s} = f_\theta(\mathbf{p})$. Thus, we denote the reasoning path of IO prompting as $\mathbf{p} \rightarrow \mathbf{s}$. As shown in Figure 3 (a), the reasoning path of IO prompting is one-step. One-step reasoning is insufficient to solve complex problems which involve multi-step reasoning.

**Chain-of-Thought Prompting**. Chain-of-Thought (CoT) Prompting (Wei et al., 2022) enables complex reasoning ability with LLMs by decomposing the reasoning path of these problems into multi-step: $\mathbf{p} \rightarrow \mathbf{z}_1 \cdots \mathbf{z}_k \rightarrow \mathbf{s}$. Here $\mathbf{z}_1 \cdots \mathbf{z}_k$ are sub-solutions in intermediate reasoning steps, a.k.a 'thoughts'. CoT uses few-shot prompts to prompt LLM to output reasoning results together with its intermediate reasoning steps: $\{\mathbf{z}_1, \cdots \mathbf{z}_k, \mathbf{s}\} = f_\theta(\mathbf{p})$. Notice this framework can be done sequentially by $\mathbf{z}_i = f_\theta(\mathbf{p}; \{\mathbf{z}_j | j < i\})$ until reaches the final solution $\mathbf{s}$ (Zhou et al., 2023).

**Tree-of-Thought Prompting**. Tree-of-Thought (ToT) Prompting formulates LLM reasoning as searching in the solution space with heuristics, such as breadth-first searching (BFS) and depth-first searching (DFS) (Yao et al., 2023). When it reaches the sub-solution $\mathbf{z}_i$ at $i$-th step, ToT employ

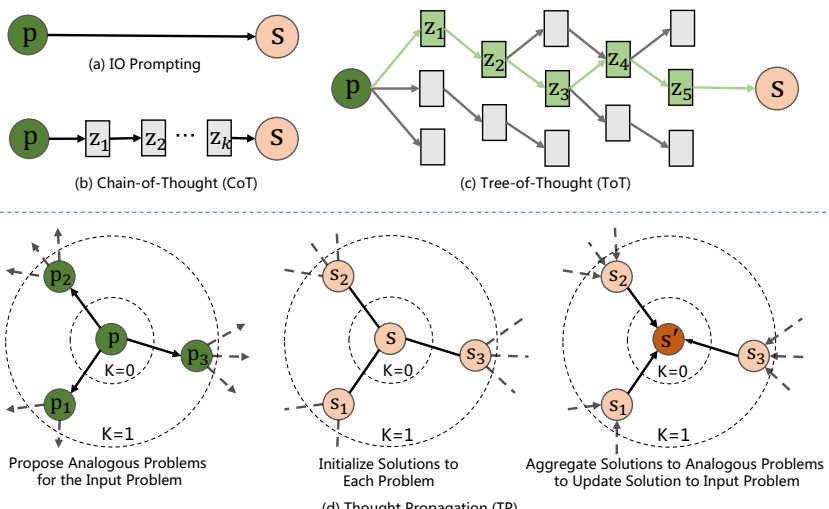

Figure 3: The illustrative comparison between Thought Propagation (TP) and other representative methods. For an input problem **p**, IO, CoT, and ToT reason from scratch to yield the solution **s**. Differently, TP explores analogous problems to improve solving the input problem.

an LLM to propose sub-solution candidates $\{\mathbf{z}_{i+1}^n | n = 1 \cdots N\} = f_\theta(\mathbf{p}; \{\mathbf{z}_j | j \leq i\})$. Then, it leverages an LLM to evaluate $\{\mathbf{z}_{i+1}^n | n = 1 \cdots N\}$ for the best one and choose it as the next sub-solution $\mathbf{z}_{i+1}$. The above searching process is repeated until ToT obtains a satisfying solution.

Although these methods improve the complex reasoning ability of LLMs with different prompting, they all prompt the LLM to reason from scratch. This reasoning scheme cannot reuse the prior knowledge in problem-solving and suffers from accumulated errors during multi-step reasoning. Thus, they fall short in tasks involving optimization and multi-step searching. As shown in Figure 2, IO, CoT, and ToT prompting all fail to find the optimal shortest path $0 \rightarrow 3 \rightarrow 4$ from Node 0 to Node 4 in a very simple graph, which can be easily solved by humans. When using multi-step searching for this task with ToT, it even falsely searches backward and visits Node 3 two times. The result of ToT on a more complex graph is shown in Figure 4 (b.2).

## 4 METHODOLOGY

When humans encounter a new problem, they often compare it to familiar ones with similar characteristics, a process known as analogical reasoning (Carbonell, 1985). Thus, we aim to leverage insights in exploring some problems that are analogous to the input problem, i.e. analogous problems, to facilitate input problem-solving. We introduce Thought Propagation (TP), a versatile analogical framework for LLM reasoning. As shown in Figure 3 (d), TP actively generates analogous problems related to the input problem during the reasoning process, all without relying on external knowledge bases. It then combines the solutions from these proposed analogous problems to facilitate solving the input problem by creating an updated solution or formulating a high-level plan. We introduce the three modules of TP: `LLM Propose`, `LLM Solve`, and `LLM Aggregate`. Then, we give a general setup of the proposed framework and leave the implementation for each task in Section 5.

**LLM Propose**. Given an input problem, `LLM Propose` generates a set of analogous problems. Solving these analogous problems should provide distinctive insights to help solve the input one. Thus, `LLM Propose` generates analogous problems in two perspectives. First, the solutions from analogous problems can be transferred to yield new solutions to the input problem. In this manner, TP can reuse the solutions from analogous problems to develop new solutions to the input problem in an analogical approach instead of reasoning from scratch. Second, solving analogous problems can produce high-level plans for the input problem-solving. In this way, TP can rectify the errors during planning and improve the multi-step reasoning of the input problem. Both ways to generate analogous problems are instantiated using few-shot problem exemplars or zero-shot prompting.

**LLM Solve**. `LLM Solve` serves a dual purpose: solving the input problem to produce an initial solution and solving the analogous problems proposed by `LLM Propose`. This module can be instantiated using existing prompting approaches such as CoT (Wei et al., 2022). Although the

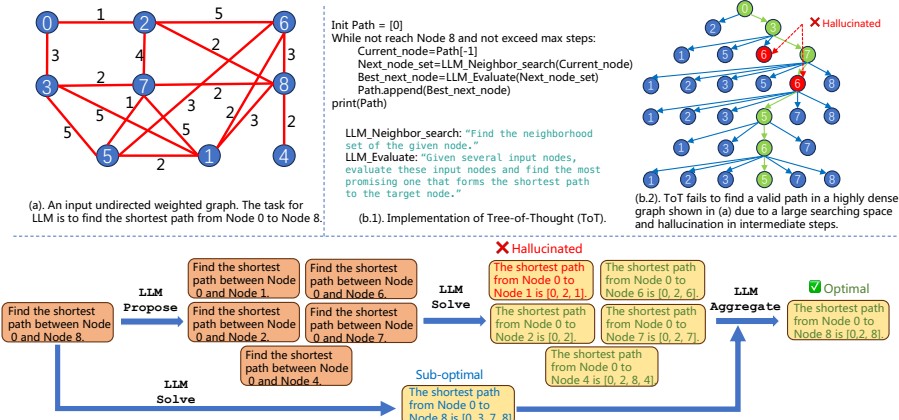

Figure 4: An example of TP and ToT for the Shortest-path Task. ToT (b) fails to solve the problem in (a) due to the accumulated errors in intermediate reasoning steps. Building upon solutions from analogous problems, TP (c) refines the initial sub-optimal solution and finally finds the optimal one.

solutions to analogous problems are not expert-level, the following aggregation module can assess these solutions and use the most promising one to instantiate analogical reasoning. Moreover, we introduce a multi-layer implementation of TP to improve solutions to analogous problems further.

**LLM Aggregate**. `LLM Aggregate` aggregates solutions from analogous problems to enhance solving the input problem. This module is conjugated to the `LLM Propose` module, since it depends on the relationship between the input problem and its analogous counterparts generated by `LLM Propose`. Thus, `LLM Aggregate` utilizes the solutions from analogous problems in two ways. First, it prompts the LLM to develop new solutions to the input problems based on the results of analogous problems. An example of this manner is shown in Figure 4. (c). If we already obtain the shortest paths to the neighborhood nodes of the target node, only one-step reasoning is required to yield a new path to the target node. Notice this manner is different from recursive problem decomposition (Khot et al., 2023) since it only requires one-step reasoning to develop a new solution to the input problem. Second, this module prompts the LLM to derive high-level plans to solve the input problem using the solutions from analogous problems. The plan is knowledge-intensive, thus the LLM can carry this plan in every round of decision-making when solving long-trial planning tasks. After generating new solutions or plans using the results of analogous problems, the LLM evaluates these outputs and chooses the best one to improve input problem-solving.

**Multi-layer Implementation**. As shown in Figure 3 (d), we can stack $K$ layers of TP to leverage the solutions from $K$-hop analogous problems to improve the solution to the input problem. Thus, existing methods, such as IO, CoT, ToT, etc., can be viewed as the special cases of TP by setting $K = 0$ since they solve each problem from scratch and do not instantiate analogical reasoning. By setting $K = 1$, TP aggregates the solutions from 1-hop analogous problems to refine the solution to the input one. TP further allows hierarchical refinement by setting $K \geq 2$. In this case, the problems in $i$-th layer are the analogous problems in $(i-1)$-th layer ($i \geq 1$). Thus, we can use the solutions from $i$-th-layer analogous problems to refine $(i-1)$-th-layer analogous problems' solutions. This hierarchical refinement finishes until the solution to the input problem is refined.

**General Setup and Recipe**. TP allows plug-and-play generalization and enhancement to different tasks, since we can use existing prompting methods for LLM reasoning to instantiate `LLM Solve`. Using IO prompting and CoT for most tasks is sufficient in our experiments. For more complex problems involving autonomous planning and exploration, prompting methods that synergize thinking and action such as ReAct (Yao et al., 2022) is needed. Although TP builds upon existing prompting methods, it aids their reasoning-from-scratch manner with the hint of solving analogous problems and leads to significant performance gain.

**Complexity Analysis**. The complexity of Thought Propagation mainly comes from two perspectives. Firstly, the complexity exponentially increases as the layer number $k$ increases. However, the $K$-hop analogous problems are intuitively less related to the input problem, and considering such long-range analogous problems only leads to marginal performance gain. Thus, we only consider implementing up to 2-layers of TP to trade off performance between complexity. We find the perfor-

Table 1: The performance of TP and other baselines on Shortest-path Reasoning Task.

| LLM-Backend | Method | 0-shot | | | 1-shot | | | 5-shot | | |
|---|---|---|---|---|---|---|---|---|---|---|
| | | OR↑ | FR↑ | OLR↓ | OR↑ | FR↑ | OLR↓ | OR↑ | FR↑ | OLR↓ |
| PaLM-2 | IO | 0.14 | 0.37 | 0.62 | 0.28 | 0.48 | 0.43 | 0.26 | 0.41 | **0.35** |
| | CoT | 0.24 | 0.52 | 0.40 | 0.33 | 0.45 | 0.41 | 0.29 | 0.56 | 0.39 |
| | BaG | 0.23 | 0.47 | 0.44 | 0.28 | 0.52 | 0.45 | 0.26 | 0.52 | 0.51 |
| | ToT | - | - | - | - | - | - | - | - | - |
| | **TP** | **0.36** | **0.57** | **0.37** | **0.38** | **0.59** | **0.36** | **0.37** | **0.62** | 0.36 |
| GPT-3.5 | IO | 0.33 | 0.50 | 0.17 | 0.62 | 0.86 | 0.15 | 0.61 | 0.9 | 0.27 |
| | CoT | 0.26 | 0.35 | 0.13 | 0.58 | 0.85 | 0.16 | 0.52 | 0.85 | 0.32 |
| | BaG | 0.25 | 0.32 | 0.13 | 0.61 | 0.87 | 0.14 | 0.64 | 0.86 | 0.13 |
| | ToT | 0.22 | 0.42 | 0.82 | 0.38 | 0.79 | 0.72 | 0.58 | 0.93 | 0.32 |
| | **TP** | **0.65** | **0.89** | **0.12** | **0.74** | **0.89** | **0.07** | **0.73** | **0.91** | **0.10** |
| GPT-4 | IO | 0.78 | 1.00 | 0.10 | 0.80 | 0.99 | 0.08 | 0.81 | 1.00 | 0.08 |
| | CoT | 0.76 | 1.00 | 0.10 | 0.75 | 1.00 | 0.11 | 0.78 | 1.00 | 0.08 |
| | BaG | 0.77 | 0.98 | 0.09 | 0.80 | 0.99 | 0.09 | 0.78 | 1.00 | 0.11 |
| | ToT | 0.46 | 0.84 | 0.52 | 0.46 | 0.73 | 0.40 | 0.77 | 1.00 | 0.07 |
| | **TP** | **0.88** | **1.00** | **0.05** | **0.88** | **1.00** | **0.04** | **0.86** | **1.00** | **0.05** |

mance gain in 2-layer TP is marginal when compared with 1-layer TP, but 2-layer TP leads to more token expenses. 1-layer TP achieves very competitive performances against the baselines with no significant increase in token expenses. For example, 1-layer TP outperforms ToT by a large margin in different LLM backends but shares similar token expenses. Secondly, instantiating `LLM Solve` under 5-shot setting is more expensive than 0-shot setting due to increasing prompting exemplars. We provide a detailed quantitative complexity analysis in Section 5.1.

# 5 EXPERIMENTS

We employ three challenging tasks, such as Shortest-Path Reasoning, Creative Writing, and LLM-Agent Planning, to evaluate the proposed method (TP). We generate 100 shortest-path problems with non-trivial solutions for the Shortest-Path Reasoning task. We employ the dataset proposed by Yao et. al. (Yao et al., 2023) with 100 writing problems for the Creative Writing task. And we use ALFWorld (Shridhar et al., 2021) game suite to instantiate the LLM-Agent Planning task with 134 environments. TP finds the most optimal shortest paths, generates the most coherent messages, and achieves the highest task completion rate in three tasks.

## 5.1 SHORTEST-PATH REASONING

The Shortest-path Reasoning task is to find the shortest path from the source node to the target node in a weighted undirected graph. This task is suitable to evaluate the complex reasoning ability of LLMs since 1. this discrete optimization problem requires searching in an explosively large space, and 2. the generated graphs in this task are not seen by LLMs to prevent data contamination.

**Task Setup**. For an input graph, LLM is required to find the shortest path from the source node to the target node using the baselines and TP. For a graph with $N$ nodes, the source node is set to Node 0, and the target node is set to Node $(N - 1)$. We filter out the trivial cases where the shortest path only contains one edge. Detailed task setup is in Appendix C.

**Baselines and LLM Backends**. We use standard (IO) prompting (Brown et al., 2020), Chain-of-Thought (CoT) (Wei et al., 2022), Build-a-Graph (BaG) (Wang et al., 2023a), and Tree-of-Thought (ToT) (Yao et al., 2023) as the baseline methods. The implementation and prompting exemplars of all the baselines are shown in Appendix C. We evaluate all the methods under 0-shot, 1-shot, and 5-shot prompting settings. We conduct experiments on three LLM backends such as PaLM 2 (Bison) (Anil et al., 2023), GPT-3.5 (OpenAI, 2022), and GPT-4 (OpenAI, 2023).

**Thought Propagation Setup**. Suppose the input problem is finding the shortest path from Node 0 to Node $(N - 1)$ in the graph $G_i$ with $N$ nodes. `LLM Propose` prompts the LLM to propose analogous problems of finding the shortest path from Node 0 to the neighborhood nodes of Node

Figure 5: Study on the complexity and performance of TP under different configurations.

$(N-1)$. `LLM Solve` is implemented with IO prompting with 0-shot/1-shot/5-shot prompting. This module outputs the initial solutions to the input problem and analogous problems. Afterward, `LLM Aggregate` uses the results of analogous problems to develop a new path to the input problem. Then, it compares the new path with the initial path and outputs a better one. The implementation and prompts are shown in Appendix C.

**Evaluation Metrics**. Denote the length of shortest path of graph $G_i$ as $L_i^*$. Let the length of the valid path output by LLM be $L_i$. $N$ is the total number of graphs. $N_{optimal}$ and $N_{feasible}$ are the number of optimal paths and valid paths output by LLMs. We propose to use three metrics to evaluate the performance of different methods in Shortest-path Reasoning. **Optimal Rate (OR)** $= N_{optimal}/N$ measures the percentage of paths generated by LLMs being the optimal paths. The higher the better. **Feasible Rate (FR)** $= N_{feasible}/N$ measures the percentage of paths generated by LLMs as the valid paths. The higher the better. **Over-Length Rate (OLR)** $= \sum_{i=1}^{N_{feasible}} \left( L_i - L_i^* \right)/L_i^*$ measures the over-length of the generated valid paths over the optimal ones. The lower the better.

**Results**. The quantitative results of TP and the baselines are shown in Table 1. TP achieves a significant performance gain over the baselines by generating the most optimal and valid shortest paths when testing on three LLM backends with different model capacities. Moreover, the valid paths generated by TP are the closest to the optimal paths when compared with the baselines due to the lowest Over-Length Rate (OLR). On the PaLM-2 backend, ToT fails to find valid paths from source nodes to target nodes. For GPT-3.5 and GPT-4 backends, ToT underperforms IO prompting. We find ToT sometimes searches backward or even fails to find the valid path due to the accumulated error shown in Figure 4. CoT only outperforms IO on PaLM-2 where IO performs badly. Nevertheless, we observe no significant preference gain of CoT over IO on the other LLM backends.

Although the 1-shot setting leads to performance gains over 0-shot on most prompting methods, the performance gains of 5-shot over 1-shot setting are unexpectedly marginal, or sometimes worse than 1-shot setting. There are two reasons for this phenomenon. Firstly, the 5-shot setting feeds long prompting exemplars to LLM, which potentially contains more redundant information. Secondly, the 5-shot setting sometimes leads to output cutoff due to the maximal token limit of LLMs. We leave the in-depth exploration of this phenomenon in our future work.

**Impact of Layers on Performance**. We further study the influence of layer numbers of TP on the complexity and performance in the Shortest-path Task. As shown in Figure 5, 1-layer TP has similar token costs as ToT in different settings. However, 1-layer TP already achieves very competitive performance in finding the optimal shortest path. Also, the performance gain of 1-layer TP over 0-layer TP (IO) is significant. Although 2-layer TP achieves the best performance as shown in Table 1, the performance gain of 2-layer TP over 1-layer TP is less significant. And Figure 5 (a). indicates the increase in the token cost of TP with 2 layers. Thus we aim to harness multi-hop analogous problems with decreased expenses in our future work. More results are shown in Appendix F.

## 5.2 CREATIVE WRITING

We proceed to evaluate Thought Propagation on the Creative Writing task (Yao et al., 2023). Given 4 randomly sampled sentences, the goal of this task is to generate 4 paragraphs ending with these sentences respectively to construct a coherent message. Such task challenges LLM reasoning by highly creative thinking and planning.

**Task Setup**. We follow the task setup proposed by Yao et. al. (Yao et al., 2023) that consists of 100 test instances. We use the coherent score (1-10 scalar score generated by GPT-4) and user study to evaluate the coherence of generated messages. The details of the evaluation are in Appendix D

Table 2: The performance of Thought Propagation (TP) and baselines on Creative Writing Task.

| Metric | Coherent Score | | User Study | |
|---|---|---|---|---|
| LLM-Backend | GPT-3.5 | GPT-4 | GPT-3.5 | GPT-4 |
| IO | $6.087 \pm 2.229$ | $6.193 \pm 1.953$ | 14% | 7% |
| CoT | $6.654 \pm 2.201$ | $6.927 \pm 1.508$ | 21% | 15% |
| ToT | $6.856 \pm 1.975$ | $7.684 \pm 1.141$ | 26% | 33% |
| **TP** | $\mathbf{7.000 \pm 1.783}$ | $\mathbf{7.989 \pm 1.453}$ | **39%** | **45%** |

Table 3: The performance of different variant models of Thought Propagation (TP) and baselines on LLM-Agent planning in the ALFWORLD dataset (Shridhar et al., 2021). We reproduce the result of Reflexion (Shinn et al., 2023). Other baseline results are quoted from ReACT (Yao et al., 2022).

| Method | Pick | Clean | Heat | Cool | Look | Pick 2 | All |
|---|---|---|---|---|---|---|---|
| BULTER | 33 | 26 | 70 | 76 | 17 | 12 | 22 |
| BULTER_G | 46 | 39 | 74 | **100** | 22 | 24 | 37 |
| Act (best of 6) | 88 | 42 | 74 | 67 | 72 | 41 | 45 |
| ReAct (avg) | 65 | 39 | 83 | 76 | 55 | 24 | 57 |
| ReAct (best of 6) | 92 | 58 | **96** | 86 | 78 | 41 | 71 |
| Reflexion | **100.00** | 74.19 | 73.91 | 85.71 | 66.67 | 70.59 | 79.1 |
| **TP-SR-SE** | **100.00** | 77.42 | 65.22 | 95.24 | **94.44** | 82.35 | 85.82 |
| **TP-SE** | 91.67 | 83.87 | 69.56 | **100.00** | 83.3 | 70.59 | 83.68 |
| **TP-SR-SM** | 95.83 | **96.77** | 78.26 | **100.00** | **94.44** | **88.24** | 92.54 |
| **TP-SM** | **100.00** | 93.55 | 86.96 | **100.00** | **94.44** | **88.24** | 94.78 |

**Baselines and LLM Backends**. We consider three baselines: IO prompting (Brown et al., 2020), CoT (Wei et al., 2022) and ToT (Yao et al., 2023). All these methods use zero-shot prompts due to the creative nature of writing (Yao et al., 2023). The baseline setup and prompting exemplars are shown in Appendix D. We instantiate each method using GPT-3.5 and GPT-4 backends.

**Thought Propagation Setup**. We build Thought Propagation with one layer for this task to maintain a fair comparison with the baselines. Every module of Thought Propagation is implemented with zero-shot prompts. `LLM Propose` rephrases the four input sentences using the simple prompt: "Rephrase the input sentences but do not change their meanings or orders.", and produces the analogical problem which is to generate a writing plan to write a message with the rephrased sentences. This module generates 5 analogous problems to ensure a fair comparison with baselines. `LLM Solve` uses CoT prompting to generate writing plans to write four paragraphs that end with four given sentences. This module is employed to solve the input and proposed analogous problems, leading to 6 plans. Since the rephrased sentences share similar contextual information with the input sentences, their writing plans potentially apply to the input ones. Thus `LLM Aggregate` evaluates all 6 plans output by `LLM Solve` and outputs the most promising plan for the input problem. Finally, the LLM is asked to write the whole message in four paragraphs using the most promising plan. The prompting exemplars of TP in the Creative Writing task are shown in Appendix D.

**Results**. Table 2 shows the performance of TP and baselines with GPT-3.5 and GPT-4. Thought Propagation outperforms the baselines with the highest coherent scores on both GPT-3.5 and GPT-4 backends. Moreover, TP achieves the highest human preference in user study. Additional findings are all the methods achieve better performance on GPT-4 due to the improved model capability.

## 5.3 LLM-AGENT PLANNING

LLM-Agents use LLMs as the core component to interact with environments and autonomously make plans and decisions. We study the capability of TP to formulate high-level, knowledge-intensive plans for LLM-Agents in an analogical way to improve the task completion rate.

**Task Setup**. ALFWorld (Shridhar et al., 2021) is a text-based game suite with various interactive housework environments aligned with ALFRED and TextWorld (Côté et al., 2019; Shridhar et al., 2020). It contains six types of tasks with 134 unseen environments for evaluation (Yao et al., 2022; Shinn et al., 2023).

**Baselines and LLM Backends**. BULTER is a trainable parameterized method based on reinforcement learning (Shridhar et al., 2021). ReAct (Yao et al., 2022) builds LLM-Agents with synergy

between reasoning traces and action trials. Act (Yao et al., 2022) removes the reasoning trace of Re-Act. Reflexion improves ReAct with verbal reflections on previous failures in the same task to refine the planning of new trials (Shinn et al., 2023). We run Reflexion for 6 trials since its performance is stable after 4 trials. We use GPT-3 for LLM-Agents following Shinn et al. (2023).

**Thought Propagation Setup**. Unlike Reflexion which reflects upon previous failure in the **same task** to help task completion in the next planning trial, Thought Propagation aims to aggregate useful information from successful trials in **similar but different tasks** to improve task completion.

Thus, `LLM Propose` uses a zero-shot prompt to assess the similarity score between the original task and the rest with successful planning trials. The rest tasks with the top two similarity scores are treated as two analogical problems.

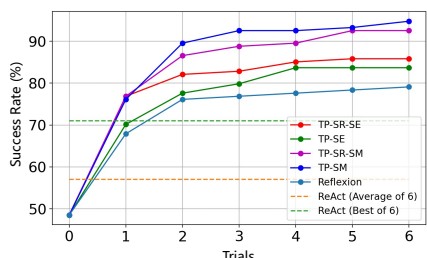

Figure 6: The success rate of task completion in different trials.

`LLM Solve` employs ReAct to instantiate LLM-Agent planning in the original task following Re-flexion (Shinn et al., 2023). `LLM Aggregate` uses a zero-shot prompt to formulate two plans to help complete the original problem based on the success-ful trials of analogical problems and the planning trial of the original problem. Then, it evaluates the two plans and outputs the better one to guide the LLM-Agent to complete the task. We run Thought Propagation for 6 trials to maintain consistency with Reflexion. The prompt exemplars are shown in Appendix E. (Shinn et al., 2023).

**Variant Models**. We introduce two strategies for `LLM Aggregate` for plan evaluation: 1. Self-Evaluation (SE): The LLM evaluates two plans by zero-shot prompt and outputs the better one; 2. Simulation (SM): The LLM-Agent executes new planning trials in the task environment using two plans and outputs a better one. We additionally add Self-Reflection (SR) modules to reflect LLM-Agent on its own failures just like Reflexion. These implementations lead to four variant models of Thought Propagation: 1). TP-SR-SE: Thought Propagation with Self-Reflection and Self-Evaluation; 2). TP-SE: Thought Propagation with Self-Evaluation; 3). TP-SR-SM: Thought Propagation with Self-Reflection and Simulation; 4). TP-SM: Thought Propagation with Simulation.

**Results**. Table 3 shows good performance of Thought Propagation over the learnable parameterized method and other LLM-Agent baselines. Thought Propagation achieves large performance gains even without a memory module to store its previous failures (TP-SE/TP-SM). This shows the supe-riority of the reflection upon successful planning in completing similar tasks. Moreover, Thought Propagation also works well with the additional memory module to store previous failures (TP-SR-SE/TP-SR-SM). Figure 6 shows that different variant models of Thought Propagation achieve consistent performance improvement by iterative reflecting on successful planning in similar tasks. We show how TP formulates a constructive plan from solving **"examine the alarmclock with the desklamp"** task to successfully complete **"examine the book with the desklamp"** task, where ReAct and Reflexion are trapped in a loop, in Appendix B.

## 6 CONCLUSIONS

Existing prompting approaches for LLM reasoning cannot leverage the insights of solving similar problems and suffer from accumulated errors in multi-step reasoning, due to reasoning from scratch. To address these issues, we propose Thought Propagation (TP), which explores analogous problems to yield a refined solution or a knowledge-intensive plan in an analogical approach to facilitate new problem-solving. TP is compatible with existing prompting methods, showing plug-and-play generalization and enhancement to a wide range of tasks such as Shortest-path Planning, Creative Writing, and LLM-Agent Planning. Future directions would further enhance the performance and efficiency of the proposed framework.

## ACKNOWLEDGEMENTS

This work is funded by the National Natural Science Foundation of China (Grant No U21B2045, U20A20223, 32341009). We appreciate the discussions with Zhuoran Yang and his suggestions.

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

## A MORE DISCUSSION WITH RELATED WORKS ON LLMS

**Retrieval-augmented LLMs.**The retrieval-augmented LLM is proposed to alleviate the hallucination phenomenon and improve the output quality of LLM (Asai et al., 2023; Mialon et al., 2023; Shi et al., 2023). For an input question, the retrieval-augmented LLM first queries an external database with billion-level tokens (Borgeaud et al., 2022; Lan et al., 2023; Zhu et al., 2023b) to retrieve a subset of text corpus to construct the output answer. The retrieval-augmented LLM achieves improved question-answering quality with fewer parameters than standard LLM (Mialon et al., 2023) and has been applied to many downstream tasks such as text/multi-modal generation (Lan et al., 2023; Yasunaga et al., 2023), question answering (Borgeaud et al., 2022; Izacard et al., 2022) and biomedical applications (Wang et al., 2023f). The retrieval-augmented LLM has been widely applied to many tasks (Lan et al., 2023; Yasunaga et al., 2023; Izacard et al., 2022; Wang et al., 2023f). Differently, the proposed method requires no external database to query from but aggregates the knowledge of solving analogous problems to hint at reasoning.

**LLM as Autonomous Agents**. LLM-Agents can interface with tools (Cai et al., 2023; Schick et al., 2023; Chen et al., 2023b; Liu et al., 2022), other LLMs (Wu et al., 2023; Li et al., 2024; Chan et al., 2023), and humans (Wang et al., 2023e; Liu et al., 2023a) to autonomously make decisions and formulate planning to solve tasks with feedback. The key component of LLM-Agents is the LLM-based planning module to process the environmental feedback and make planning (Wang et al., 2023b; Zhu et al., 2023a). When deployed in long-trial decisions and planning scenarios, LLM-Agents should make multiple rounds of action and planning. As LLMs are likely to hallucinate, LLM-Agents will accumulate errors and finally fail to complete planning tasks (Yao et al., 2022). Recent work amends this issue by reflecting on its previous failures in completing the same task when LLM-Agents start one more try for task completion (Shinn et al., 2023). Differently, we apply the proposed method to help LLM-Agents summarize their successful experience in completing other similar tasks and formulate a plan to improve task completion.

## B EXAMPLES OF LONG-TRIAL LLM-AGENT PLANNING

In this section, we provide an example to show how TP can generate analogous problems, formulate a plan from these results, and further facilitate solving original planning tasks in the ALFworld dataset (Shridhar et al., 2021). Notice that this example is not cherry-pick. For the following task, the LLM-Agent implemented with ReAct (Yao et al., 2022) fails to complete the task of examining the book with the desklamp and is trapped in a loop.

> Here is the task: You are in the middle of a room. Looking quickly around you, you see a bed 1, a desk 2, a desk 1, a drawer 6, a drawer 5, a drawer 4, a drawer 3, a drawer 2, a drawer 1, a garbagecan 1, a laundryhamper 1, a safe 1, a shelf 6, a shelf 5, a shelf 4, a shelf 3, a shelf 2, and a shelf 1. Your task is to: examine the book with the desklamp.
> Agent think: To solve the task, I need to find and take a book, then find and use a desklamp. First I need to find a book. A book is more likely to appear in desk (1-2), shelf (1-6), bed (1), garbagecan (1), drawer (1-6), laundryhamper (1). I can check one by one, starting with desk 1.
> Feedback OK.
> Agent
> Feedback Nothing happens.
> Agent go to desk 1
> Feedback On the desk 1, you see a alarmclock 2, a creditcard 1, a desklamp 1, and a keychain 2.
> Agent think: Now I find a desklamp (1). Next, I need to take it.
> Feedback OK.
> Agent take desklamp 1 from desk 1
> Feedback Nothing happens.
> Agent take desklamp 1 from desk 1
> Feedback Nothing happens.
> Agent take desklamp 1 from desk 1

> Feedback Nothing happens.
> STATUS: FAIL

Reflexion (Shinn et al., 2023) proposed a memory system that stores the failure memory of LLM-Agent on previous planning trials in trying to finish the same task. However, it is still trapped in a loop of trying to take the desklamp instead of finding a book and fails to complete this task.

> Your memory for the task below:
> Trial 0:
> I will go to shelf 3, take desklamp 1, then examine the book with the desklamp. If I am stuck in a loop again, I will try to execute a different action.
> Trial 1:
> I was stuck in a loop in which I continually tried to take desklamp 1 from shelf 3 instead of examining the book with the desklamp. I should have gone to desk 1, found the desklamp, then looked for the book. I will go to desk 1, find the lamp, then look for the book. If I am stuck in a loop again, I will try to execute a different action.
> Trial 2:
> I will go to desk 1, find the desklamp, then look for the book. I will take the book from desk 2, then examine it with the desklamp. If I am stuck in a loop again, I will try to execute a different action.
> Here is the task:
> You are in the middle of a room. Looking quickly around you, you see a bed 1, a desk 2, a desk 1, a drawer 6, a drawer 5, a drawer 4, a drawer 3, a drawer 2, a drawer 1, a garbagecan 1, a laundryhamper 1, a safe 1, a shelf 6, a shelf 5, a shelf 4, a shelf 3, a shelf 2, and a shelf 1.
> Your task is to: examine the book with the desklamp.
> Agent think: To solve the task, I need to find and take a book, then find and use a desklamp. First I need to find a book. A book is more likely to appear in desk (1-2), shelf (1-6), bed (1), garbagecan (1), drawer (1-6), laundryhamper (1). I can check one by one, starting with desk 1.
> Feedback OK.
> Agent
> Feedback Nothing happens.
> Agent go to desk 1
> Feedback On the desk 1, you see a alarmclock 2, a creditcard 1, a desklamp 1, and a keychain 2.
> Agent think: Now I find a desklamp (1). Next, I need to take it.
> Feedback OK.
> Agent take desklamp 1 from desk 1
> Feedback Nothing happens.
> Agent take desklamp 1 from desk 1
> Feedback Nothing happens.
> STATUS: FAIL

TP first employs the `LLM Propose` to propose 2 analogous problems with task indexes 129 and 6. Then it looks at the planning trials in the last round of task completion and uses `LLM Aggregate` to summarize these two plans to assist the LLM-Agent to complete the original task.

> Plans from successful attempts in similar tasks:
> Plan #0: I should have started by checking the drawers (1-6) first, then the desk (1-2), shelf (1-6), bed (1), garbagecan (1), and laundryhamper (1). I should have checked each one in order, starting with drawer 1. I should have opened each drawer and checked the contents, and if nothing was found, I should have moved on to the next one. I should have checked the desk (1-2) next, and then the shelf (1-6). If nothing was found, I should have checked the bed (1), garbagecan (1), and laundryhamper (1). If nothing was found, I should have gone back to the drawers (1-6) and checked them again.
> Plans from successful attempts in similar tasks:
> Plan #1: To solve the task, I need to find and take a book, then find and use a desklamp. First I need to find a book. A book is more likely to appear in desk (1-2), shelf (1-6), bed (1), garbagecan (1), drawer (1-6), laundryhamper (1). I can check one by one, starting with desk 1. I will check desk 1 and desk 2, then shelf 1-6, then bed 1, then garbagecan 1, then drawer 1-6, and finally laundryhamper 1. If I don't find a book in any of these locations, I will go back to the beginning and start again.

These two plans are more detailed and instructive than those generated and stored in the memory module of Reflexion (Shinn et al., 2023) by reflecting on its previous failures in completing the **same** task. Then, `LLM Aggregate` further evaluates these two plans with simulation and outputs the better plan: Plan #1 obtained by summarizing the planning trial of task with index 6. The planning trial of task with index 6 is as follows:

> Here is the task: You are in the middle of a room. Looking quickly around you, you see a bed 1, a desk 2, a desk 1, a drawer 6, a drawer 5, a drawer 4, a drawer 3, a drawer 2, a drawer 1, a garbagecan 1, a laundryhamper 1, a safe 1, a shelf 6, a shelf 5, a shelf 4, a shelf 3, a shelf 2, and a shelf 1.
> Your task is to: examine the alarmclock with the desklamp.
> Agent think: To solve the task, I need to find and take an alarmclock, then find and use a desklamp. First I need to find an alarmclock. An alarmclock is more likely to appear in desk (1-2), drawer (1-6), shelf (1-6), bed (1), garbagecan (1), laundryhamper (1). I can check one by one, starting with desk 1.
> Feedback OK.
> Agent go to desk 1
> Feedback On the desk 1, you see a bowl 1, a desklamp 1, a mug 1, and a pencil 1.
> Agent go to desk 2
> Feedback On the desk 2, you see a alarmclock 1.
> Agent think: Now I find an alarmclock (1). Next, I need to take it.
> Feedback OK.
> Agent take alarmclock 1 from desk 2
> Feedback You pick up the alarmclock 1 from the desk 2.
> Agent think: Now I take an alarmclock (1). Next, I need to find a desklamp. A desklamp is more likely to appear in desk (1-2), shelf (1-6), bed (1), drawer (1-6), garbagecan (1), laundryhamper (1). I can check one by one, starting with desk 1.
> Feedback OK.
> Agent go to desk 1
> Feedback On the desk 1, you see a bowl 1, a desklamp 1, a mug 1, and a pencil 1.
> Agent think: Now I find a desklamp (1). Next, I need to use it.
> Feedback OK.
> Agent use desklamp 1
> Feedback You turn on the desklamp 1.
> STATUS: OK

This task is **to examine the alarmclock with the desklamp**. It is very intuitive that the experience and skill developed in solving the **examine the alarmclock with the desklamp** problem can be reused and further facilitate solving **examine the book with the desklamp** due to the similarity of both tasks. It is interesting that TP autonomously formulates the constructive and knowledge-intensive plan to help solve **examine the book with the desklamp**. The LLM-Agent then carries

this plan in each decision step to run a new planning trial and successfully complete this task as follows. Notice that the generated plan from analogous problems successfully guides the LLM-Agent to run a long-trial planning and successfully finish the task.

---

Plans from successful attempts in similar tasks:

Plan #1: To solve the task, I need to find and take a book, then find and use a desklamp. First I need to find a book. A book is more likely to appear in desk (1-2), shelf (1-6), bed (1), garbagecan (1), drawer (1-6), laundryhamper (1). I can check one by one, starting with desk 1. I will check desk 1 and desk 2, then shelf 1-6, then bed 1, then garbagecan 1, then drawer 1-6, and finally laundryhamper 1. If I don't find a book in any of these locations, I will go back to the beginning and start again.

Here is the task:

You are in the middle of a room. Looking quickly around you, you see a bed 1, a desk 2, a desk 1, a drawer 6, a drawer 5, a drawer 4, a drawer 3, a drawer 2, a drawer 1, a garbagecan 1, a laundryhamper 1, a safe 1, a shelf 6, a shelf 5, a shelf 4, a shelf 3, a shelf 2, and a shelf 1.

Your task is to: examine the book with the desklamp.

Agent think: To solve the task, I need to find and take a book, then find and use a desklamp. First I need to find a book. A book is more likely to appear in desk (1-2), shelf (1-6), bed (1), garbagecan (1), drawer (1-6), laundryhamper (1). I can check one by one, starting with desk 1.

Feedback OK.

Agent go to desk 1

Feedback On the desk 1, you see a alarmclock 2, a creditcard 1, a desklamp 1, and a keychain 2.

Agent go to desk 2

Feedback On the desk 2, you see a alarmclock 3, a bowl 1, a creditcard 3, a creditcard 2, a mug 3, a mug 2, and a mug 1.

Agent go to shelf 1

Feedback On the shelf 1, you see a alarmclock 1, and a pen 1.

Agent go to shelf 2

Feedback On the shelf 2, you see a pencil 1.

Agent go to shelf 3

Feedback On the shelf 3, you see a alarmclock 2, and a desklamp 1.

Agent go to shelf 4

Feedback On the shelf 4, you see nothing.

Agent go to shelf 5

Feedback On the shelf 5, you see nothing.

Agent go to shelf 6

Feedback On the shelf 6, you see nothing.

Agent go to bed 1

Feedback On the bed 1, you see a book 1, a laptop 1, a pillow 2, and a pillow 1.

Agent think: Now I find a book (1). Next, I need to take it.

Feedback OK.

Agent take book 1 from bed 1

Feedback You pick up the book 1 from the bed 1.

Agent think: Now I take a book (1). Next, I need to find a desklamp. A desklamp is more likely to appear in desk (1-2), shelf (1-6), bed (1), garbagecan (1), drawer (1-6), laundryhamper (1). I can check one by one, starting with desk 1.

Feedback OK.

Agent go to desk 1

Feedback On the desk 1, you see a alarmclock 2, a creditcard 1, a desklamp 1, and a keychain 2.

Agent think: Now I find a desklamp (1). Next, I need to use it.

Feedback OK.

Agent use desklamp 1

Feedback You turn on the desklamp 1.

---

> STATUS: OK

## C  PROMPT EXAMPLE FOR SHORTEST-PATH REASONING

### C.1  TASK SETUP

We consider the shortest path problem on graphs. LLM is required to find the shortest path from the source node to the target node using different prompt-based methods. We generate 100 undirected weighted graphs with diverse node numbers and edge connections. The number of nodes is randomly sampled between 5 and 10. The graph generation algorithm is similar to generating an Erdős-Rényi random graph but forces the generated graphs to be connected. We follow the following code originally from `https://www.fil.univ-lille.fr/~varre/portail/ graphes/tp/TP4.html` for graph generation and implement it using Python. We set $p = 0.2$ to avoid from generating sparse graphs. After generating the graphs, we randomly sample their edge weights from 1 to 5 as their edge distances. The graphs are transformed into sequences of node lists, edge lists, and edge distance lists, which are fed into the LLMs. For a graph with $N$ nodes, the task is to find the shortest path from Node 0 to Node $(N-1)$.

```
def gnp_random_connected_graph(n, p):

    edges = combinations(range(n), 2)
    G = nx.Graph()
    G.add_nodes_from(range(n))
    if p <= 0:
        return G
    if p >= 1:
        return nx.complete_graph(n, create_using=G)
    for _, node_edges in groupby(edges, key=lambda x: x[0]):
        node_edges = list(node_edges)
        random_edge = random.choice(node_edges)
        G.add_edge(*random_edge)
        for e in node_edges:
            if random.random() < p:
                G.add_edge(*e)
    return G
```

### C.2  BASELINES AND LLM BACKENDS

We use standard (IO) prompting (Brown et al., 2020), Chain-of-Thought (CoT) (Wei et al., 2022), Build-a-Graph (BaG) Wang et al. (2023a), and Tree-of-Thought (ToT) (Yao et al., 2023). IO prompting uses several pairs of input graphs and their shortest paths as prompting exemplars. CoT employs the input graphs and their shortest paths with intermediate reasoning steps of the form "Starting from Node $i$, we reach Node $j$. The distance between two nodes is." BaG is a recently proposed method to enhance LLM's reasoning performance on graphs by asking the LLM to build a graph first. We re-implement ToT as a breadth-first searching (BFS) starts from the source node. When reaching an intermediate node, ToT first proposes its neighbor nodes and evaluates them for the best one that forms the shortest path to the target node. We evaluate all the methods under zero-shot, one-shot, and five-shot prompting settings. We conduct experiments on three LLM backends such as PaLM 2 (Anil et al., 2023), GPT-3.5 (OpenAI, 2022), and GPT-4 (OpenAI, 2023).

### C.3  PROMPTING EXAMPLES FOR THOUGHT PROPAGATION AND BASELINES

We convert the input graphs into lists of node sets, edge sets, and edge distance sets to feed a graph into LLMs for shortest-path reasoning. We set the source node and the target node by appending them to the end of the above lists. We provide a prompting exemplar of IO prompting.

> Find the shortest path from a source node to a target node in an undirected graph. The undirected graph is represented as a node set, an edge set, and an edge distance set.
> Input:
> Node set: [0, 1, 2, 3, 4]
> Edge set: [[0, 3], [1, 4], [2, 4], [3, 4]]
> Edge distance set: [2, 3, 5, 3]
> Source Node: 0
> Target Node: 4
> Answer: The shortest path from the source node to the target node is [0, 3, 4]. The shortest distance is 5.
> Input:
> {input}

The 5-shot setting contains 5 input examples with their answers using the above format. For the 0-shot setting, we just set the input and output format based on the above prompting exemplar but do not use any specific shortest path example as the prompting exemplar.

> Find the shortest path from a source node to a target node in an undirected graph. The undirected graph is represented as a node set, an edge set, and an edge distance set.
> The format of the input and answer are below:
> Input:
> Node set: []
> Edge set: []
> Edge distance set: []
> Source Node: source node index
> Target Node: target node index
> Answer:
> The shortest path from the source node to the target node is [source node index, ..., target node index]. The shortest distance is BLANK.
> Input:
> {input}

The CoT prompting for shortest-path reasoning decomposes the short-path reasoning into finding the nodes in the optimal shortest path one by one. The 5-shot prompting exemplars are similar to IO prompting. The 0-shot prompting exemplars add a "Let's think step by step" following the zero-shot CoT (Kojima et al., 2022) and we just provide the 1-shot prompting as follows.

> Find the shortest path from a source node to a target node in an undirected graph. The undirected graph is represented as a node set, an edge set, and an edge distance set.
> Input:
> Node set: [0, 1, 2, 3, 4]
> Edge set: [[0, 3], [1, 4], [2, 4], [3, 4]]
> Edge distance set: [2, 3, 5, 3]
> Source Node: 0
> Target Node: 4
> Answer:
> Starting from node 0, we arrive at node 3. The distance between these two nodes is 2.
> Starting from node 3, we arrive at node 4. The distance between these two nodes is 3.
> Thus, the shortest path from the source node to the target node is [0, 3, 4]. The shortest distance is 5.
> Input:
> {input}

Since Build-a-Graph (BaG) Qian et al. (2023) prompting has no publicly available implementation yet, we follow the implementation in its preprint paper as follows.

> Find the shortest path from a source node to a target node in an undirected graph. The undirected graph is represented as a node set, an edge set, and an edge distance set. Let's construct a graph with the nodes and edges first.
> Input:
> Node set: [0, 1, 2, 3, 4]
> Edge set: [[0, 3], [1, 4], [2, 4], [3, 4]]
> Edge distance set: [2, 3, 5, 3]
> Source Node: 0
> Target Node: 4
> Answer:
> Starting from node 0, we arrive at node 3. The distance between these two nodes is 2.
> Starting from node 3, we arrive at node 4. The distance between these two nodes is 3.
> Thus, the shortest path from the source node to the target node is [0, 3, 4]. The shortest distance is 5.
> Input:
> {input}

For ToT Yao et al. (2023), we implement its node evaluation module to find the most promising node that forms the shortest path to the target node. 0-shot, 1-shot, and 5-shot settings of ToT contain 0, 1, 4 input examples respectively.

> Given several input nodes, evaluate these input nodes and find the most promising one that forms the shortest path to the target node.
> Input:
> Node set: [0, 1, 2, 3, 4, 5, 6]
> Edge set: [[0, 1], [0, 3], [0, 4], [1, 4], [2, 4], [3, 4], [2, 5], [2, 6]]
> Edge distance set: [2, 2, 2, 3, 3, 5, 4, 1]
> Input Nodes: [3, 4]
> Target Node: 6
> Answer:
> The most promising one that forms the shortest path to the target node in the input nodes is 4. The shortest path is [4, 2, 6]. The shortest distance is 4.
> Input:
> {input}

The neighborhood proposing module of ToT is the same as Thought Propagation. This module proposes several neighborhood nodes of the current node in the ToT searching steps. We use 5-shot diverse prompting examples to instantiate this module for both ToT and Thought Propagation.

The `LLM Solve` module of Thought Propagation is implemented using 0-shot, 1-shot, and 5-shot IO prompting. The `LLM Propose` module is as follows:

> The undirected graph is represented as a node set, an edge set. Given an input node, find its neighborhood nodes and save them in a list.
> Input:
> Node set: [0, 1, 2, 3, 4]
> Edge set: [[0, 3], [1, 4], [2, 4], [3, 4]]
> Input Node: 4
> Answer:
> The neighborhood node list of the input node is [1, 2, 3].
> Input:
> {input}

After finding the neighborhood nodes of the target node, `LLM Propose` module uses these neighborhood nodes to generate several analogous problems. For neighborhood node Node $i$, `LLM Propose` generates the analogous problem in the format of "Find the shortest path from Node

0 to Node i", which is solved by `LLM Solve`. After solving all the analogous problems, `LLM Aggregate` uses the following prompts to aggregate the solutions from analogous problems to yield a new solution to the original problem.

The undirected graph is represented as a node set, an edge set and an edge distance set. The edges in the edge set are reversible. We have hints of one or several intermediate paths from the source node to some intermediate nodes. Please use these hints to find the shortest path from the source node the the target node.
Input:
Node set: [0, 1, 2, 3, 4]
Edge set: [[0, 3], [1, 4], [2, 4], [3, 4]]
Edge distance set: [2, 3, 5, 3]
The hints are:
The shortest path from the source node 0 to the intermediate node 3 is [0, 3]. The shortest distance is 2.
The shortest path from the source node 0 to the intermediate node 1 is [0, 3, 4, 1]. The shortest distance is 8.
The shortest path from the source node 0 to the intermediate node 2 is [0, 3, 4, 2]. The shortest distance is 10.
Use the above hint to find the shortest path from the source node 0 to the target node 4.
Answer:
Using the above hints, the shortest path from the source node 0 to the target node 4 is [0, 3, 4]. The shortest distance is 5.
Input:
{input}

After generate a new solution of shortest path problem, `LLM Aggregate` use the following prompt to evaluate the new solution and the initial solution and output a better one.

The undirected graph is represented as a node set, an edge set and an edge distance set. The edges in the edge set are reversible. We have two solution candidates for the shortest path from the source node to the target node. Please verify these two solution candidates and output the better one. If two solutions are the same and both valid, output the first solution. Notice that the shortest distance provided in the hints may be wrong. Check it before using it.
Input:
Node set: [0, 1, 2, 3, 4, 5]
Edge set: [[0, 3], [0, 2], [1, 3], [1, 5], [2, 3], [2, 5], [3, 4], [4, 5]]
Edge distance set: [2, 2, 5, 4, 2, 3, 3, 4]
Source Node: 0
Target Node: 5
Solution 1: The shortest path from the source node 0 to the target node 5 is [0, 2, 5]. The shortest distance is 5.
Solution 2: The shortest path from the source node 0 to the target node 5 is [0, 3, 4, 5]. The shortest distance is 9.
Answer:
Solution 1 is valid because it can reach the target node and all the edges in Solution 1 are real edges in the Edge set. Solution 2 is valid because it can reach the target node and all the edges in Solution 2 are real edges in the Edge set. Solution 1 is better than Solution 2 because the path in Solution 1 is shorter than that in Solution 2. So the shortest path from the source node 0 to the target node 5 is [0, 2, 5]. The shortest distance is 5.
Input:
{input}

# D  PROMPT EXAMPLE FOR CREATIVE WRITING

## D.1  TASK SETUP

We follow the task setup proposed by Yao et. al. (Yao et al., 2023). We use a dataset consisting of 100 test instances. Each test instance contains 4 sentences which are randomly sampled from this website [1]. We use the coherent score and the user study to evaluate the coherence of generated messages. For the coherent score, we employ GPT-4 with a zero-shot prompt to give a 1-10 scalar score to evaluate the coherence of the generated message. GPT-4 evaluates each message 5 times and outputs the average score. For user study, we employ the lab members excluding the paper authors to compare the coherence of the generated messages between Thought Propagation and other baselines. The order of messages is flipped in the user study.

## D.2  BASELINES AND LLM BACKENDS

We consider three baselines: IO prompting (Brown et al., 2020), CoT (Wei et al., 2022) and ToT Yao et al. (2023). All these methods use zero-shot prompts due to the creative nature of writing (Yao et al., 2023). IO prompts the LLM to generate a coherent message given the sentences. CoT first prompts the LLM to formulate a writing plan to elicit intermediate reasoning and generate the whole message using the plan. Both IO and CoT generate 10 samples for each test instance. We implement ToT with one intermediate thought step following Yao et. al. (Yao et al., 2023). ToT first prompts the LLM to generate 5 writing plans and vote for the best one. Then, it generates the whole message 5 times and votes for the best one as the output. We follow the implementations of IO, CoT, ToT in Yao et. al. (Yao et al., 2023).

We introduce the implementation of each module of Thought Propagation. `LLM Solve` generates the plans of writing four paragraphs.

> Make a writing plan for a coherent passage of 4 short paragraphs. The end sentence of each paragraph must be:
> {input}
> The plan contains a one-sentence description in each paragraph. Your output should be in the following format:
> Plan:
> Your plan here.

The `LLM Propose` rephrases the input 4 sentences but does not change their order or meaning to construct an analogous problem. The rephrased sentences are sent to `LLM Solve` to generate a new writing plan.

> Please rephrase the input sentences but do not change their order or meaning. The input sentences are:
> {input}
> Your output should be in the following format:
> Output:
> Your sentences here.

We follow the scoring prompt in ToT (Yao et al., 2023) to instantiate `LLM Aggregate` and evaluate the initial plan with the new writing plans to output the best one.

> Given several writing plans, decide which writing plan is the most promising. Analyze each writing plan in detail, then conclude in the last line "The best choice is s", where s is the integer id of the choice.

Finally, TP uses the following prompt to writing 4 paragraphs using the best plan.

---

[1] https://randomwordgenerator.com/sentence.php

> Following this plan: {plan} to write a coherent passage of 4 short paragraphs. The end sentence of each paragraph must be: {input}
> Your output should be in the following format:
> Passage:
> Your passage here.

# E    PROMPT EXAMPLE FOR LLM-AGENT PLANNING

## E.1    TASK SETUP

ALFWorld (Shridhar et al., 2021) is a text-based game suite with various interactive housework environments aligned with ALFRED and TextWorld (Côté et al., 2019; Shridhar et al., 2020). It contains six task categories such as picking up objects, finding objects, manipulating objects, and so on. To complete these tasks, the LLM-Agent needs to interact with the environment to obtain feedback and autonomously make multi-step decisions and planning. We use 134 unseen environments for evaluating different methods following (Yao et al., 2022; Shinn et al., 2023). The expense of running 6 trials of TP on this dataset is about 300-400 dollars, which is similar to Reflexion.

## E.2    PROMPTING EXEMPLAR OF TP IN LLM-AGENT PLANNING

We follow the experiment setting of Reflexion (Shinn et al., 2023). To instantiate `LLM Solve`, we use ReAct with the same prompting in Reflexion. `LLM Propose` use the following prompt to evaluate the input problem with the successfully solved ones in the last round of trials. The two problems with the highest scores are treated as analogous problems.

> Evaluate the similarity score between these two cases. The similarity score should be an integer between 0 and 10. 0 indicates the least similarity and 10 indicates the most similarity.
> Case 1:
> {current task description}
> Case 2:
> {proposed task description}
> The output format should be: The similarity score is:

For `LLM Aggregate`, it first uses the planning trials of successfully solved analogous problems proposed by `LLM Propose` to devise two plans to help with input problem completion.

> You will be given a successful case where you successfully complete the task. Then you will be given a failure case where you fail to complete the task. Do not summarize these two cases, but rather use the successful case to think about the strategy and path you took to attempt to complete the task in the failure case. Devise a concise, new plan of action that accounts for your mistake with reference to specific actions that you should have taken. For example, if you tried A and B but forgot C, then devise a plan to achieve C with environment-specific actions. You will need this later to solve the failure case. Give your plan after "Plan".
> Success Case:
> {success case}
> Failure Case:
> {failure case}
> Plan:

`LLM Aggregate` evaluate two plans and output the better one. The better plan is sent to LLM-Agent to carry with to solve the task in the next round of planning trials.

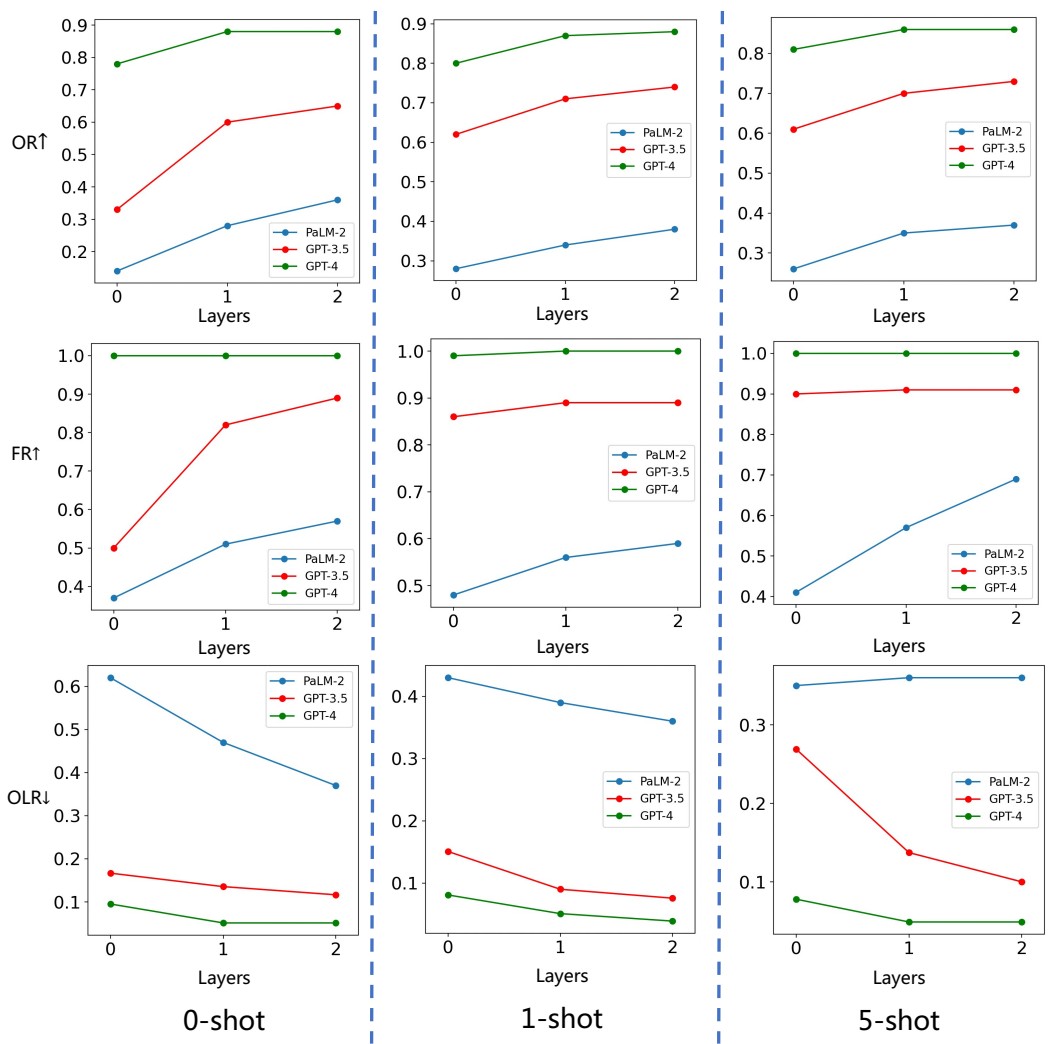

Figure 7: The influence of layer number on the performance of Thought Propagation on Shortest-path Reasoning.

> You once fail to accomplish the following task.
> The task is: {task}
> Here are several plans to accomplish the task:
> {several plans}
> Output the most promising plan to accomplish the task, but do not output any irrelevant messages. Your answer goes after Plan:
> Plan:

## F    IMPACT OF LAYERS ON PERFORMANCE

We study the impact of different layer numbers on the model performance in Figure 7.

## G    MORE DISCUSSION WITH SELF-REFINEMENT OF LLM REASONING

The self-refinement, also known as self-critic, is an emerging technique to rectify the mistakes and hallucinations during the inference time of LLM reasoning (Madaan et al., 2024). The intuition

behind self-refinement is to extend the fundamental ability of humans, who improve their previous decisions based on self-feedback or external feedback, to the reasoning process of LLMs. The early attempt of self-refinement of LLM reasoning employs a two-stage manner (Huang et al., 2022). First, a pre-trained LLM employs Chain-of-Thought (Wei et al., 2022) with Self-Consistency (Wang et al., 2023d) to output the reasoning results with high confidence for a dataset of reasoning problems. Then, the reasoning problems with their results are leveraged to fine-tune the pre-trained LLM for improved performances in reasoning. Recent works on self-refinement resort to prompting methods to examine the correctness of LLM reasoning output with internal (Madaan et al., 2024; Shinn et al., 2023) or external feedback (Welleck et al., 2022; Ganguli et al., 2023) , and further refine the results if needed. The internal feedback comes from the same LLM that performs reasoning simultaneously (Shinn et al., 2023). The external feedback is generated by other LLMs as critics (Wang et al., 2023c; Du et al., 2023; Gao et al., 2023), tools for interaction (Gou et al., 2023; Chen et al., 2023b), and humans Saunders et al. (2022); Ouyang et al. (2022).

The `LLM Aggregate` module of the proposed Thought Propagation (TP) shares a similar motivation with the self-refinement methods by improving the solution to the input problem (please see more details in Section 4). However, existing self-refinement methods still handle each input problem separately, without reusing the experience of solving analogous problems to refine the input problem-solving just like TP. This limits the existing method not to exploring diverse strategies for refinement and thus usually leads to unsatisfying refinement. As shown in Figure 6, all variant models of the proposed TP achieve significantly higher task completion rate than Reflexion (Shinn et al., 2023), which is a representative self-refinement method by reflecting upon its previous failures in completing the ***same*** task. Also, the performance gains of TP over the baselines are significant in the other tasks as shown in Section 5, thanks to the advantages of teaching LLMs to think analogically. Thus, LLMs can explore a wide range of strategies by reusing the insights of handling analogous problems, and assess them for the most promising one for refinement. As shown in the example in Appendix B, only TP yields an effective strategy for the LLM-Agent to refine its previous planning while other methods generate ineffective strategies that trap the LLM-Agent in a loop.

## H    MORE EMPIRICAL ANALYSIS ON SHORTEST-PATH REASONING TASKS

### H.1    IMPLEMENTING THOUGHT PROPAGATION WITH CHAIN-OF-THOUGHT

The proposed Thought Propagation (TP) is a plug-and-play analogical reasoning approach to enhance the complex reasoning ability of LLMs. We further instantiate the `LLM Solve` module with Chain-of-Thought (Wei et al., 2022) (CoT) prompting and rerun the new model, namely TP+CoT. In addition, we implement one and two layers of TP propagation for TP+CoT, leading to two variant models: TP (1-layer)+CoT and TP (2-layer)+CoT. We rerun CoT with GPT 3.5 to ensure a fair comparison with two variant models due to the changing behavior of the GPT model (Chen et al., 2023a). Thus, the obtained performance of CoT is slightly different from that in Table 1. As shown in Table 4, TP (1 Layer)+CoT and TP (2 Layer)+CoT outperform CoT with significant performance gains. This indicates that TP enjoys plug-and-play enhancement for different prompting methods.

### H.2    IMPACT OF DIFFERENT GRAPH ENCODING METHODS ON THE MODEL PERFORMANCE IN SHORTEST-PATH REASONING TASK

We encode the input graph-structured data into sequential data, which consists of node lists, edge lists, and edge distance lists since LLMs can only consume sequential data. The reasons behind this graph encoding method, namely Adjacency, are twofold. Firstly, graphs are usually represented as the collection of node lists, edge lists, and edge attribute lists. In our case, the edge attributes are edge distances. Thus, it is straightforward to convert the input graphs into the sequences of node lists, edge lists, and edge distance lists. Secondly, the obtained sequential data can preserve the graph connectivity information.

Previous works have shown that changing the prompt exemplars can impact the reasoning performances of prompting methods. As there are multiple ways to encode the input graphs into sequential data, we further study the influence of different graph encoding methods on the performances of prompting approaches. Specifically, we consider two additional graph encoding methods including Edge Description (Wang et al., 2023a) and Graph Modeling Language (Guo et al., 2023;

Table 4: The comparison between Thought Propagation (TP) with different layers when implemented with IO prompting and Chain-of-Thought (CoT). We evaluate all the models with GPT 3.5. The performance of IO, TP (1 Layer)+IO, and TP (2 Layer)+IO are copied from Section 5. We rerun CoT to ensure a fair comparison with TP (1 Layer)+CoT and TP (2 Layer)+CoT due to the changing behavior of the GPT model (Chen et al., 2023a).

| Method | 0-shot | | | 1-shot | | | 5-shot | | |
|---|---|---|---|---|---|---|---|---|---|
| | OR↑ | FR↑ | OLR↓ | OR↑ | FR↑ | OLR↓ | OR↑ | FR↑ | OLR↓ |
| IO | 0.33 | 0.50 | 0.17 | 0.62 | 0.86 | 0.15 | 0.61 | 0.90 | 0.27 |
| TP (1 Layer)+IO | 0.60 | 0.82 | 0.14 | 0.71 | **0.89** | 0.09 | 0.70 | **0.91** | 0.14 |
| TP (2 Layer)+IO | **0.65** | **0.89** | **0.12** | **0.74** | 0.89 | **0.07** | **0.73** | **0.91** | **0.10** |
| CoT | 0.30 | 0.42 | **0.14** | 0.62 | 0.87 | 0.15 | 0.63 | 0.97 | 0.20 |
| CoT-SC | 0.46 | 0.55 | 0.17 | 0.67 | 0.9 | 0.13 | 0.66 | 0.97 | 0.18 |
| TP (1 Layer)+CoT | 0.53 | 0.75 | 0.19 | 0.68 | **0.91** | 0.11 | 0.72 | **0.98** | 0.15 |
| TP (2 Layer)+CoT | **0.60** | **0.83** | 0.18 | **0.71** | 0.89 | **0.09** | **0.73** | 0.96 | **0.13** |

Table 5: The performance comparison between methods with different graph encoding. We run all the experiments using GPT-3.5.

| Graph Encoding | Method | 0-shot | | | 1-shot | | | 5-shot | | |
|---|---|---|---|---|---|---|---|---|---|---|
| | | OR↑ | FR↑ | OLR↓ | OR↑ | FR↑ | OLR↓ | OR↑ | FR↑ | OLR↓ |
| Adjacency | IO | 0.33 | 0.50 | 0.17 | 0.62 | 0.86 | 0.15 | 0.61 | 0.90 | 0.27 |
| | CoT | 0.26 | 0.35 | 0.13 | 0.58 | 0.85 | 0.16 | 0.52 | 0.85 | 0.32 |
| | BoG | 0.25 | 0.32 | 0.13 | 0.61 | 0.87 | 0.14 | 0.64 | 0.86 | 0.13 |
| | ToT | 0.22 | 0.42 | 0.82 | 0.38 | 0.79 | 0.72 | 0.58 | 0.93 | 0.32 |
| | TP | **0.65** | **0.89** | **0.12** | **0.74** | **0.89** | **0.07** | **0.73** | **0.91** | **0.10** |
| Edge Description | IO | 0.72 | 0.95 | 0.14 | 0.70 | 0.97 | 0.19 | 0.70 | 0.95 | 0.16 |
| | CoT | 0.67 | 0.84 | 0.15 | 0.74 | 0.97 | 0.15 | 0.71 | **0.97** | 0.16 |
| | BoG | 0.63 | 0.85 | 0.13 | 0.73 | 0.98 | 0.13 | 0.70 | 0.94 | 0.12 |
| | ToT | 0.12 | 0.27 | 0.92 | 0.23 | 0.47 | 0.82 | 0.57 | **0.97** | 0.49 |
| | TP | **0.80** | **0.97** | **0.09** | **0.83** | **0.99** | **0.09** | **0.80** | 0.94 | **0.09** |
| Graph Modeling Language | IO | 0.73 | **0.99** | 0.12 | 0.75 | 0.92 | 0.09 | 0.70 | 0.91 | 0.08 |
| | CoT | 0.40 | 0.47 | 0.05 | 0.72 | **0.93** | 0.10 | 0.70 | **0.98** | 0.13 |
| | BoG | 0.72 | 0.97 | 0.10 | 0.73 | 0.92 | 0.10 | 0.71 | 0.90 | 0.10 |
| | ToT | 0.13 | 0.18 | 0.85 | 0.21 | 0.44 | 1.04 | 0.49 | 0.84 | 0.48 |
| | TP | **0.86** | **0.99** | **0.04** | **0.84** | **0.93** | **0.02** | **0.79** | 0.93 | **0.05** |

Himsolt, 1997). For an input graph with $N$ nodes, Edge Description represents the input graph as *"In an undirected graph, the nodes are numbered from 0 to N, and the edges are represented as: an edge between node $i$ and node $j$ with distance $LENGTH$, ···".* Similarly, Graph Modeling Language converts the input graph into the following sequence *"graph[comment "This is an undirected graph." node [id 0] ··· node [id N] edge [label "Edge between node $i$ and node $j$ with distance $LENGTH$"] ···.]"*

As shown in Table 5, different graph encoding methods lead to the change in performances of prompting methods in the Shortest-path Reasoning task. For example, Edge Description and Graph Modeling Language both improve the reasoning ability of most prompting methods under 0-shot and few-shot settings when compared with Adjacency. This indicates that these two graph encoding methods help LLMs to better understand the input graph structures.

Although different approaches to graph encoding indeed impact the reasoning performances of the prompting methods, our work does not focus on finding the best way to encode graph input into sequences. Instead, our work aims to develop a general analogical approach to enhance the complex reasoning ability of LLMs. The proposed TP also enjoys an over 10% absolute increase in finding the optimal shortest path when adopting Edge Description and Graph Modeling Language to encode graphs into sequences as shown in Table 5. Moreover, TP also demonstrates significant performance gains in the other two tasks in Section 5.

