# OpenReview forum: "THOUGHT PROPAGATION: AN ANALOGICAL APPROACH TO COMPLEX REASONING WITH LARGE LANGUAGE MODELS"
_ICLR.cc/2024/Conference — ICLR 2024 poster_

### Official Review · Reviewer_yu29 · 2023-11-01

**Soundness:** 3 good
**Presentation:** 2 fair
**Contribution:** 3 good
**Rating:** 6
**Confidence:** 4

**Summary:**

This paper proposes a new LLM prompting strategy, Thought Propagation (TP), which generates analogies to the input problem, generates solutions to them, evaluates the solutions (using the LLM itself), and then uses the correct solutions either to directly solve the input problem, or to derive a high-level plan to solve the input problem. TP is evaluated against relevant baseline methods on three tasks:  shortest-path reasoning, creative writing, and planning in ALFWorld. The results show that TP consistently obtains higher scores than the baseline methods tested.

**Strengths:**

The surprising and potentially exciting finding is that this reasoning-through-generated-analogies helps solve the input problem even without ground-truth validation of the correctness of the solutions to the analogies. It's surprising because of the counterintuitive nature of the finding; one might have expected ToT to outperform TP since ToT spends its compute on the input problem rather than on analogies which are, by definition, somewhat different than the input problem.

The work deals with an important topic, is carefully motivated, and the results are analyzed in detail.

**Weaknesses:**

Since TP outperforms ToT by a surprising and counterintuitive amount, it is critical to compare the two methods in terms of token cost, which is a controllable quantity for both TP and ToT. The authors acknowledge the importance of comparing token cost in their discussion of Figure 5:  “1-layer TP outperforms ToT by a large margin in different LLM backends but shares similar token expenses.” However, this figure applies only to the shortest-path task, which is highly problematic for evaluating LLMs. The task is contrived in the sense that one would never actually use an LLM to solve a shortest-path task, and one would not expect much LLM training data to be relevant to solving such tasks. Even more importantly, the results of this evaluation are likely to depend very sensitively on the details of how the graph definition is represented in natural language for insertion into the prompt. Therefore one would expect that making subtly different choices in representation could make very large differences in performance, and some of those choices would just happen to favor certain prompt approaches (like TP) over others (like ToT). For these reasons, it seems hard to conclude much from the shortest-path experiments. Unfortunately, no token costs are provided for the other two experiments. This leads me to doubt the validity of the paper’s central finding.

**Post-Rebuttal Comments**

I commend the authors for the extra experiments providing token-level comparisons between prompting techniques on the non-graph tasks. Other additional experiments explored different ways of encoding graphs, and TP maintained advantages over ToT. For these reasons, I am raising my evaluation from 5 to 6.

I still question the relevance of the shortest-path experiments. In rebuttal, the authors argued that the shortest-path task is a good testbed for evaluating LLM reasoning for 3 reasons:
- It is a challenging task, requiring many steps to solve over a variety of graphs.
- Some concurrent work has applied LLMs to graphs.
- Generated graphs avoid data contamination.

I agree that application of LLMs to graphs is an interesting line of investigation, where these 3 reasons make sense. But the goal of this work, as staked out in the title and abstract, is to compare different prompting techniques across a broader range of LLM applications. If the paper’s claims were confined to the graph domain, the significance of the work would be greatly limited.

**Questions:**

None

---

> ### Author Response · Authors · 2023-11-19
> **Official Response by Authors (1/2)**
>
> Thank you for acknowledging the motivation and novelty of our work! And we really appreciate your valuable suggestions and comments. We would like to answer the questions in Weaknesses as follows:
>
> Q1: The reasons why we evaluate LLMs on Shortest-path Reasoning Task.
>
> A1: Thanks for this valuable comment. The reasons are as follows:
>
> - $\textbf{The Shortest-path Reasoning Task is a good testbed for evaluating the complex reasoning ability of LLMs.}$ Although humans have invented deterministic algorithms to solve the shortest-path reasoning task, this task is suitable to test the complex reasoning ability of LLMs. Unlike the arithmetic problems in prior works requiring a few steps to solve, the shortest-path task takes many more steps to solve and requires LLMs to traverse in discrete, diverse, and irregular graphs for optimal solutions. Thus, this task poses great challenges for LLMs. Moreover, some concurrent works also start to explore the ability of LLMs to solve graph-related tasks [1,2], since LLMs have achieved great success in many applications. These works indicate that it is difficult for LLMs to solve graph-related tasks. Thus, the shortest-path task serves as a good testbed to evaluate the complex reasoning ability of LLMs.
>
> - $\textbf{Using generated graphs for evaluation prevents data contamination.}$ Recent works highlight the risk in LLM evaluation that the testing data are used to train LLMs, which is known as data contamination [4]. However, the graph modality is different from the sequential data used to train LLMs. Moreover, we randomly generate diverse graphs for LLM evaluation with the algorithm introduced in Appendix C.1. The LLMs such as GPT-3.5/4.0 cannot use our generated graphs, which were generated in July 2023, and were never uploaded to the Internet, for parameter training. Thus, evaluating LLMs on the Shortest-path Reasoning task with generated graphs can prevent data contamination. We will clarify the motivation for evaluating LLMs with the Shortest-path Reasoning task in the Experiment part of the revision.
>
> Q2: The impact of how graphs are encoded into prompts in natural languages may affect the performances of different methods.
>
> A2: Thanks for this valuable comment and we have added a detailed analysis in Appendix H.2! In our work, we initially adopted the straightforward manner to encode input graphs into the collection of node lists, edge lists, and edge distance lists. We denote such graph encoding method as $\textbf{Adjacency}$.
>
> Previous works have shown that changing the prompt exemplars can impact the reasoning performances of prompting methods. As there are multiple ways to encode the input graphs into sequential data, we further consider two additional graph encoding methods including $\textbf{Edge Description}$ [1] and $\textbf{Graph Modeling Language}$ [2,3].
>
> For an input graph with $N$ nodes, Edge Description represents the input graph as "In an undirected graph, the nodes are numbered from 0 to $N$, and the edges are represented as: an edge between node $i$ and node $j$ with distance $LENGTH$,$\cdots$.". Graph Modeling Language converts the input graph into the following sequence "graph[comment "This is an undirected graph." node [id 0] $\cdots$ node [id $N$] edge [label "Edge between node $i$ and node $j$ with distance $LENGTH$"] $\cdots$.]"
>
> As shown in Table 1 below, different graph encoding methods indeed impact the performances of different prompting approaches. However, the proposed TP still enjoys an over 10\% absolute increase in finding the optimal shortest path when adopting Edge Description and Graph Modeling Language to encode graphs into sequences. $\textbf{This indicates the robustness of TP against different graph encoding methods.}$
>
> Notice that we do not aim to propose the best way for LLMs to understand graphs. Instead, we aim to propose a general analogical approach to enhance the complex reasoning ability of LLMs across different tasks. As discussed in Section 5, TP achieves significant performance gains in different tasks not limited to Shortest-path reasoning.

---

> ### Author Response · Authors · 2023-11-19
> **Official Response by Authors (2/2)**
>
> [Table 1: The performance comparison between methods with different graph encoding. We run all the experiments using GPT-3.5.]
> | Graph Encoding | Method | 0-shot OR↑ | 0-shot FR↑ | 0-shot OLR↓ | 1-shot OR↑ | 1-shot FR↑ | 1-shot OLR↓ | 5-shot OR↑ | 5-shot FR↑ | 5-shot OLR↓ |
> |----------------|--------|------------|------------|-------------|------------|------------|-------------|------------|------------|-------------|
> | Adjacency      | IO     | 0.33       | 0.50       | 0.17        | 0.62       | 0.86       | 0.15        | 0.61       | 0.90       | 0.27        |
> |                | CoT    | 0.26       | 0.35       | 0.13        | 0.58       | 0.85       | 0.16        | 0.52       | 0.85       | 0.32        |
> |                | BoG    | 0.25       | 0.32       | 0.13        | 0.61       | 0.87       | 0.14        | 0.64       | 0.86       | 0.13        |
> |                | ToT    | 0.22       | 0.42       | 0.82        | 0.38       | 0.79       | 0.72        | 0.58       | 0.93       | 0.32        |
> |                | TP     | **0.65**   | **0.89**   | **0.12**    | **0.74**   | **0.89**   | **0.07**    | **0.73**   | **0.91**   | **0.10**    |
> | Edge           | IO     | 0.72       | 0.95       | 0.14        | 0.70       | 0.97       | 0.19        | 0.70       | 0.95       | 0.16        |
> | Description    | CoT    | 0.67       | 0.84       | 0.15        | 0.74       | 0.97       | 0.15        | 0.71       | **0.97**   | 0.16        |
> |                | BoG    | 0.63       | 0.85       | 0.13        | 0.73       | 0.98       | 0.13        | 0.70       | 0.94       | 0.12        |
> |                | ToT    | 0.12       | 0.27       | 0.92        | 0.23       | 0.47       | 0.82        | 0.57       | **0.97**   | 0.49        |
> |                | TP     | **0.80**   | **0.97**   | **0.09**    | **0.83**   | **0.99**   | **0.09**    | **0.80**   | 0.94       | **0.09**    |
> | Graph          | IO     | 0.73       | **0.99**   | 0.12        | 0.75       | 0.92       | 0.09        | 0.70       | 0.91       | 0.08        |
> | Modeling       | CoT    | 0.40       | 0.47       | 0.05        | 0.72       | **0.93**   | 0.10        | 0.70       | **0.98**   | 0.13        |
> | Language       | BoG    | 0.72       | 0.97       | 0.10        | 0.73       | 0.92       | 0.10        | 0.71       | 0.90       | 0.10        |
> |                | ToT    | 0.13       | 0.18       | 0.85        | 0.21       | 0.44       | 1.04        | 0.49       | 0.84       | 0.48        |
> |                | TP     | **0.86**   | **0.99**   | **0.04**    | **0.84**   | **0.93**   | **0.02**    | **0.79**   | 0.93       | **0.05**    |
>
> Q3: The Computation cost analysis on Creative writing and LLM-agent Planning tasks.
>
> A3: Thanks for this valuable comments! We agree that TP is more computationally expensive than ToT as TP needs to explore a set of analogous problems to update the initial input problem-solving. ToT uses 790.1K tokens for Creative writing while TP uses 2869.1K tokens. For the LLM-agent Planning task, it is challenging to accurately calculate the number of tokens due to some dependency issues of ALFWORLD. However, we managed to roughly estimate that Reflexion costs around 300 USD to run 6 trails while the different variant models of TP cost 300-400 USD to run 6 trails based on the billing information from an LLM research institute/company.
>
> To address the concerns that TP achieves better performances than baselines by simply using more tokens, we rerun CoT 10 times and max-vote the best results (CoT-SC [5]). Please refer to A2 in response to Reviewer 39VY for detailed experiment results. Our finding is that 1-layer TP+IO already outperforms CoT-SC. This indicates the efficacy of the proposed TP results from the novel analogical reasoning approach rather than only increasing the token cost. We will aim to further improve the efficacy of the proposed TP while reducing its computation cost in our future work. One solution is to incorporate the task-related knowledge in \texttt{LLM Propose} module. Thus, TP could propose the analogous problems that most help the input problem-solving. And we will add the above analysis to the Experiment part of the final version.
>
> [1]. Can Language Models Solve Graph Problems in Natural Language? NeurIPS 2023.
> [2]. Gml: A portable graph file format. Technical report, Universitat Passau, 1997.
> [3]. Gpt4graph: Can large language models understand graph-structured data? an empirical evaluation and benchmarking. Arxiv 2023.
> [4]. Data Contamination: From Memorization to Exploitation. ACL 2022.
> [5]. Self-consistency improves the chain of thought reasoning in language models. ICLR 2023.

---

### Official Review · Reviewer_39VY · 2023-11-01

**Soundness:** 3 good
**Presentation:** 3 good
**Contribution:** 3 good
**Rating:** 6
**Confidence:** 4

**Summary:**

This text introduces Thought Propagation (TP), an approach to enhance Large Language Models' (LLMs) reasoning abilities. TP leverages insights from solving analogous problems to improve complex reasoning. It prompts LLMs to propose and solve related analogous problems, reusing their solutions and problem-solving strategies. Experiments show that, TP outperforms existing methods such as Chain-of-Thought and Tree-of-Thought on three challenging tasks by large margins.

**Strengths:**

1. The idea that exploring analogous problems and leveraging the solutions to prompt the reasoning task of LLMs is interesting and with broad interests to the research and application of LLMs. The authors provide many analyses and examples to show the efficacy of the proposed thought propagation (TP), which are convicing.

2. The method does not require to train the LLMs with sophisticated strategy or careful design of datasets in some previous works, but is a plug-and-play approach in inference, which is efficient and environment friendly, and can be generalize to various LLMs.

3. The performance improvements are significant.

**Weaknesses:**

1. TP is training-free, but compared to train-of-thought, it requires more action steps to propose, solve, and aggregate analogous problems, which is more computationally-costly and complex.

2. Some typos: in page 9, the last sentence of section 6, the quotation marks should be `` ''.

**Questions:**

1. Can we combine TP with CoT to achieve further performance improvements?

---

> ### Author Response · Authors · 2023-11-19
> **Official Response by Authors**
>
> Thank you for acknowledging the novelty and contributions of our work! And we indeed appreciate the valuable feedback from Reviewer 39VY. We hope the following responses could address your concerns.
>
> Q1: Combining TP with CoT to achieve further performance improvements.
>
> A1: Thanks very much for the valuable suggestion! And we have added an additional section in Appendix H.1 for detailed analysis. We adopt CoT to instantiate the $\texttt{LLM Solve}$ module of TP and test the obtained model, namely TP+CoT, on the Shortest-path Reasoning Task. In addition, we implement one and two layers of TP propagation for TP+CoT, leading to two variant models: TP (1-layer)+CoT and TP (2-layer)+CoT. We rerun CoT with GPT-3.5-turbo to ensure a fair comparison with two variant models due to the changing behavior of the GPT model [1]. Thus, the obtained performance of CoT is slightly different from that in our manuscript. We use the 3 metrics, namely Optimal Rate (OR↑), Feasible Rate (FR↑), and Over-Length Rate (OLR↓), as introduced in Section 5.1 for evaluation. We run all the models under 0-shot, 1-shot, and 5-shot settings.
>
> [Table 1: Comparison between IO, CoT, CoT-SC, TP+IO, and TP+CoT in the Shortest-path Reasoning task.]
> | Method            | 0-shot OR↑ | 0-shot FR↑ | 0-shot OLR↓ | 1-shot OR↑ | 1-shot FR↑ | 1-shot OLR↓ | 5-shot OR↑ | 5-shot FR↑ | 5-shot OLR↓ |
> |-------------------|------------|------------|-------------|------------|------------|-------------|------------|------------|-------------|
> | IO                | 0.33       | 0.50       | 0.17        | 0.62       | 0.86       | 0.15        | 0.61       | 0.90       | 0.27        |
> | TP (1 Layer)+IO   | 0.60       | 0.82       | 0.14        | 0.71       | **0.89**   | 0.09        | 0.70       | **0.91**   | 0.14        |
> | TP (2 Layer)+IO   | **0.65**   | **0.89**   | **0.12**    | **0.74**   | **0.89**   | **0.07**    | **0.73**   | **0.91**   | **0.10**    |
> | CoT               | 0.30       | 0.42       | **0.14**    | 0.62       | 0.87       | 0.15        | 0.63       | 0.97       | 0.20        |
> |CoT-SC(n=10) | 0.46	|0.55	|0.17	|0.67	|0.90	|0.13	|0.66	|0.97	|0.18
> | TP (1 Layer)+CoT  | 0.53       | 0.75       | 0.19        | 0.68       | **0.91**   | 0.11        | 0.72       | **0.98**   | 0.15        |
> | TP (2 Layer)+CoT  | **0.60**   | **0.83**   | 0.18        | **0.71**   | 0.89       | **0.09**    | **0.73**   | 0.96       | **0.13**    |
>
>
> As shown in Table 1 above, TP (1 Layer)+CoT and TP (2 Layer)+CoT outperform CoT with significant performance gains, especially under the 0-shot setting. This indicates that TP enjoys plug-and-play enhancement for different prompting methods. Moreover, we find CoT and TP+CoT could not beat their IO prompting counterparts in the Shortest-path Reasoning task under 0-shot and few-shot settings, which is consistent with our findings and analysis in Section 5.1.
>
> Q2: Increased complexity when compared with training methods and chain-of-thought (CoT) prompting.
>
> A2: Thanks for your valuable comments! As TP is a prompting method, it is free from the burdensome of collecting additional data and training parameters of neural nets. Thus, it is efficient, generalizable, and eco-friendly when compared with training methods.
>
> However, TP indeed requires additional computation costs to instantiate analogical reasoning when compared with reason-from-scratch prompting methods such as chain-of-thought (CoT) prompting. As introduced in Section 4, the additional computation cost mainly comes from solving an increasing number of analogous problems when implementing $k$ layer of TP. Thus, we only set $k=1$ or $k=2$ as they already lead to significant performance gain over the baselines. One solution to decrease the computational cost of TP is to further incorporate the task-related knowledge in $\texttt{LLM Propose}$ module to propose the most `helpful' analogous problems for the input problem-solving. We leave exploration in this direction in our future work, as we focus on a general analogical reasoning approach in this work.
>
> Additionally, we study whether CoT with self-consistency (CoT-SC [2]), which solves the input problems $n$ times and votes for the most promising solutions, can outperform TP when these two approaches share similar token costs. We set $n=10$ to ensure they share similar token costs. As shown in Table 1, 1-layer TP+IO already outperforms CoT-SC. This indicates the efficacy of the proposed TP benefits from the novel analogical reasoning approach rather than only increasing the token cost.
>
> Q3: Typo issue in the last sentence of Section 6 on Page 9.
>
> A3: We really appreciate your suggestions and we have corrected the quotation marks in the last sentence of section 6 on page 9.
>
>
> [1]. How is ChatGPT's behavior changing over time? Arxiv 2023.
> [2]. Self-consistency improves the chain of thought reasoning in language models. ICLR 2023.

---

### Official Review · Reviewer_VZuc · 2023-11-03

**Soundness:** 3 good
**Presentation:** 3 good
**Contribution:** 3 good
**Rating:** 8
**Confidence:** 4

**Summary:**

Current prompt-based methods limit LLMs from using past knowledge, making complex tasks difficult. This work alleviates this issue and improving the reasoning performance of LLMs on complex problems with a novel Thought Propagation (TP) framework, which is rooted in the fundamental analogical reasoning ability of human cognition. Given an input problem, TP prompts LLMs to seek for its analogous problems. Then, it initializes the solutions to the input problem and its analogous counterparts with existing prompting methods such as standard prompt, CoT etc. Finally, TP instantiates analogical reasoning to update the initial solution to the input problem in two ways: 1. directly develops a refined solution to the input problems and 2. devise a knowledge-intensive plan to improve input problem solving. All these steps are automated with LLMs by prompting. Thus, TP teach LLMs to reason in an analogical way and achieves plug-and-play enhancement to current prompt methods without extensive labor in task-specific prompt engineering. Experiments on three challenging tasks validate the generality and significant performance gain of the proposed TP framework.

**Strengths:**

* Originality: This work introduces the novel TP framework to enhance LLMs' reasoning by reusing experience in solving similar problems. While previous research explored analogical reasoning on knowledge graphs, a generalized analogical reasoning framework for LLMs was missing until TP. It includes three automated modules: LLM Propose, LLM Solve, and LLM Aggregate, providing a plug-and-play advantage over existing "reason-from-scratch" methods. TP also extends the success of neighborhood propagation in Graph Neural Networks to analogical problem-solving, an innovative and non-trivial generalization, sparking new directions in LLM reasoning.

* Clarity: This paper is overall well-structured and easy to follow. The general setup in the section of methodology helps readers to understand the proposed TP framework. Additionally, the detailed information on TP for task instantiation in the Experiment and Appendix sections facilitates implementation and reproducibility.

* Evaluation: Extensive evaluation across three tasks demonstrates TP's substantial performance improvement compared to baseline methods across various LLM backbones.

* Significance: TP's modular design exhibits impressive generality across various tasks. It has great chances to benefit researches in diverse directions.

**Weaknesses:**

Authors should enhance the related work section for a more thorough comparison.

* The authors are encouraged to compare TP with Self-refined LLM reasoning methods [1,2] since TP also manages to refine the solution to the input problems in LLM Aggregation module.
* In shortest path reasoning tasks, does TP sometimes deteriorate the solutions to some testing instances instead of improving them?

[1] Self-Refine: Iterative Refinement with Self-Feedback.

[2] Large Language Models Can Self-Improve.

**Questions:**

Please refer to weaknesses part.

---

> ### Author Response · Authors · 2023-11-19
> **Official Response by Authors**
>
> We greatly thank Reviewer VZuc for acknowledging the novelty and contribution of our work! We have carefully revised our submission according to your constructive suggestions. The responses to your questions are as follows:
>
> Q1: Compared with self-refinement LLM reasoning methods.
>
> A1: We thank the reviewer for bringing our attention to these works. We have added an additional section in Appendix G (Pages 25-26) to compare the proposed work (TP) with self-refined LLM reasoning in detail and highlight our contributions. The self-refined LLM reasoning, also known as self-critic, is an emerging technique to rectify the mistakes and hallucinations during the inference time of LLM reasoning [1,2]. Recent works in self-refined LLM reasoning guide LLMs to examine the correctness of their output with internal [3] or external feedback [4], and further refine their output if needed.
>
> The $\texttt{LLM Aggregate}$ module of TP shares similar intuition with self-refined LLM reasoning methods, since it can yield strategies for improving the input problem-solving with the insights of solving $\textbf{analogous}$ problems. By analogical reasoning, $\texttt{LLM Aggregate}$ can assess a diverse set of strategies and choose the best one for refinement based on the connection between the input problem and its analogous counterparts. Instead, existing self-refined LLM reasoning methods cannot utilize the knowledge of solving analogous problems to refine the input problems as they cannot prompt LLMs to perform analogical reasoning. We will explore the application of TP in self-refined LLM reasoning in the future.
>
> Q2: The probability of deteriorating the initial solution after Thought Propagation.
>
> A2: Thank you very much for this question! We further divide the cases where TP changes the initial solutions to input problems by analogical reasoning into 2 categories:
>
> - Improved: the percentage of cases where TP changes the invalid solutions to valid ones and improves the valid solutions to better ones,
>
> - Deteriorated: the percentage of cases where TP changes the valid solutions into invalid ones and changes the valid solutions to worse (but also valid) ones.
>
> We compute the probability of these two categories under 0-shot, 1-shot, and 5-shot in the Shortest-path Reasoning task. The results are in the following table.
>
> |                |          |      0-shot       |             |          |       1-shot      |             |          |       5-shot      |             |
> |----------------|---------------|-------------|-------------|---------------|-------------|-------------|---------------|-------------|-------------|
> | Backends       | Palm-2        | GPT-3.5     | GPT-4       | Palm-2        | GPT-3.5     | GPT-4       | Palm-2        | GPT-3.5     | GPT-4       |
> | Improved       | 68.97%         | 85.71%      | 86.96%      | 63.77%         | 83.72%      | 73.98%      | 71.74%         | 88.95%      | 100%        |
> | Deteriorated   | 31.03%         | 14.29%      | 13.04%      | 36.23%         | 16.28%      | 26.02%      | 28.26%         | 11.05%      | 0%          |
>
> As shown in the table above, TP improves more initial solutions in the Shortest-path Reasoning task rather than deteriorates them. This indicates that TP can greatly refine the initial solutions, which are obtained by reasoning from scratch, to enhance the complex reasoning ability of LLMs by analogical reasoning. Moreover, we observe that TP is more capable of improving initial solutions when deployed on more powerful LLMs such as GPT series. In our future work, we aim to further improve the efficacy of the TP framework and delve into the theoretical analysis behind analogical reasoning with LLMs.
>
> [1]. Self-refine: Iterative refinement with self-feedback. NeurIPS 2023.
> [2]. Large Language Models can self-improve. Arxiv 2022.
> [3]. Reflexion: an autonomous agent with dynamic memory and self-reflection. NeurIPS 2023.
> [4]. The capacity for moral self-correction in large language models. Arxiv 2023.

---

### Author Response · Authors · 2023-11-19
**General Responses to All Reviewers and ACs.**

Dear Reviewers and ACs,

We would like to appreciate the time and effort that all the reviewers and ACs spent during reviewing. Your constructive comments have significantly contributed to enhancing the quality of our submission. We appreciate the positive feedback (Reviewer VZuc, 39VY, and yu29) that acknowledges the motivation, novelty, and contributions of the proposed Thought Propagation (TP). In addition, all reviewers recognize the good performances of TP in various tasks through extensive experiments.

We hope the questions and concerns from all the reviewers could be addressed by the thorough literature review (Reviewer VZuc), additional empirical comparison (Reviewer 39VY), and more experimental analysis (Reviewer VZuc and yu29). We also update our submission according to the suggestions from all the reviewers. Please contact us if you have further questions.

Best,
Authors

---

### Meta-Review · Area_Chair_GWLx · 2023-12-14

**Metareview:**

This paper proposes a new approach called Thought Propagation (TP) for leveraging LLMs for solving reasoning tasks. TP takes three sequential steps: (1) prompting LLMs to first generate multiple problems similar to the target problem; (2) prompting LLMs for solving the similar, generated problems and then (3)  using LLMs to combine the generated answers to create a final answer to the target problem.

TP is shown to outperform state-of-the-art methods (Tree of Thoughts and Chain of Thoughts) on most tested benchmarks. However, the downside raised by some reviewers (`39VY`, `yu29`) and that the authors agree to is that TP is substantially slower than ToT and CoT.

The authors did a satisfactory job in answering the rebuttal questions from all three reviewers and requests for additional results.

I vote for the paper to be `Accepted` given its extensive experiments, large quantitative improvement (especially in 0-shot and lower-shot settings) compared to previous methods.
However, I'd request the authors to include the computation details/disadvantages in the Appendix and refer to it from a Limitation discussion in the main text.

**Justification For Why Not Higher Score:**

Despite being more accurate, TP is also substantially more computationally expensive compared to prior methods.

**Justification For Why Not Lower Score:**

The paper has extensive experiments showing its superior accuracy compared to baselines.

---

### Decision · Program_Chairs · 2024-01-16

Accept (poster)